# Chicken swarm optimization modelling for cognitive radio networks using deep belief network-enabled spectrum sensing technique

**Saraswathi M.**[ORCID]*, **Logashanmugam E.**

Department of Electronics and Communication Engineering, Sathyabama Institute of Science and Technology, Chennai, Tamil Nadu, India

* msaraswathi.ece@gmail.com

**Data Availability Statement:** All relevant data are within the manuscript.

**Funding:** The author received no specific funding for this work.

## Abstract

Cognitive radio networks (CRN) enable wireless devices to sense the radio spectrum, determine the frequency state channels, and reconfigure the communication variables to satisfy Quality of Service (QoS) needs by reducing energy utilization. In CRN, spectrum sensing is an essential process that is highly challenging and can be addressed by several traditional techniques, such as energy detection, match filtering, etc. For now, the current models' performance is impacted by the comparatively low Signal to Noise Ratio (SNR) of recognized signals and the insignificant quantity of traditional signal samples. This research proposals a new spectral sensing technique for cognitive radio networks (SST-CRN) that addresses the drawbacks of predictable energy detection models. With the use of a deep belief network (DBN), the suggested model contributes to accomplish a nonlinear threshold based on the chicken swarm algorithm (CSA). The proposed DBN enabled SST-CRN technique goes through two phases in a organized process: offline and online. Throughout the offline phase, the DBN model is methodically trained on pre-gathered data, developing the aptitude to identify problematic patterns and examples from the spectral features of the radio environment. This stage involves extensive feature extraction, validation, and model development to ensure that the DBN can professionally represent complicated spectral dynamics. Additionally, online spectrum sensing is conducted during the real communication phase to enable real-time adaptation to dynamic changes in the spectrum environment. Offline spectrum sensing is typically performed during a devoted sensing period before actual communication begins. When combined with DBN's deep learning capabilities and CSO's innate nature-inspired algorithms, a synergistic framework is created that enables CRNs to explore and allocate incidences on their own with astonishing accuracy. The proposed solution considerably improves the spectrum efficiency and resilience of CRNs by harnessing the power of DBN, which leads to more effective resource utilization and less interference. The Simulation results show that our proposed strategy produces more accurate spectrum occupancy assessments. The result parameters such as probability of detection, SNR of -24dB, the SST-CRN perfect has increased a developed Pd of 0.810, whereas the existing methods RMLSSCRN-100 and RMLSSCRN-300 have accomplished a lower Pd of 0.577 and 0.736, respectively. Our deep learning methodology uses convolutional neural networks to

**Competing interests:** The authors have declared that no competing interests exist.

automatically learn and adapt to dynamic and complicated radio environments, improving accuracy and flexibility over classic spectrum sensing approaches. Future research might focus on improving CSO algorithms to better optimize the spectrum sensing process, enhancing the reliability of DBN-enabled sensing techniques.

## 1. Introduction

The radio spectrum is not readily available because to the continuously expanding number of wireless devices and the stationary way in which it is organized [1]. Therefore, one of the most significant issues facing future communication research that still has to be solved is the scarcity of radio spectrum. Using cognitive radio technologies [2] is one way to deal with this issue as well as others. These technologies enable a wireless expedient to sense the radio spectrum, determine the frequency station's status, and modify its transmission parameters to minimize energy consumption while sustaining quality of service requirements [3]. This device may employ approved and unlicensed bands while their authorized primary user isn't active, preventing adverse interference.

Because wireless channels are time-varying, shadowing, and fading, spectrum sensing (SS) has been a important challenge for cognitive radio technology [4]. Many SS methods, such as cyclo-stationary-based detecting, matched filtering, wavelet-based sensing, waveform-based sensing, energy detection sensing, and eigenvalue-based sensing, were introduced in the research for the determination of sensing new or limited incidence bands. Several sensing devices that fall within the wideband and narrowband categories have recently been introduced. While wideband sensing examines several frequencies at once, narrowband sensing only appearances at one frequency station at a time [5]. Cyclo-stationary Feature Detection (CFD) is the technique of recognizing and evaluating cyclo-stationary characteristics in a signal. Certain types of signals exhibit cyclo-stationarity, which means that their statistical features change periodically across time. Energy Detection (ED) is commonly used for SS due to its noncoherent approach, which does not need previous knowledge of PU signals for the SU receiver to operate. Its sensing time is fast and its processing complexity is low. In wireless communication systems, noise uncertainty a random and inevitable fluctuation in noise significantly reduces ED performance, mainly at low SNR.

The spectrum is typically divided into numerous subbands, which can then be sensed either successively or concurrently using narrowband sensing methods. A number of spectrum sensing (SS) techniques have been proposed in current decades, and they can be categorized as narrowband or wideband depending on the bandwidth that is crucial for spectrum sensing; wideband spectrum sensing analyzes multiple frequencies concurrently, while narrowband spectrum sensing only appearances at one incidence band at a time. Sequential sensing methods could be more efficient since they need higher energy and longer times because of Analog to Digital converter (ADC), i.e., impractical and costly for timely communication. A simultaneous sensing scheme requires many sensor nodes, and a joint synchronized function increases the complication of a particular performance [6].

Energy detection is a SS model that depends on determining the absence/presence of the primary user and measuring the established signal energy by relating the selected energy level with inception. Extensive research was conducted in the survey to find the optimum threshold expression and also to develop SS performances [7]. In [8], the researchers projected a novel technique for adaptive threshold selection in multi-band detection. Evaluating the threshold can be implemented by the function of the primary and secondary indicators of the established signals. The detection performance could be evaluated using two metrics: detection probability

(DP) signifies the chance that a CR user will report a PU is available when the spectra are really being used by a PU, and false alarm probability (FA) represents the chance that a CR user will report a PU is accessible while the spectra are blank. For best detection performance, it is usually necessary when the probability of detection is gradually vulnerable to an FA, as a detection miss would require PU intervention and an FA would decrease the SE [9].

A potential research gap in the existing method of SST-CRN approach based on the DBN-Enabled spectrum sensing technique is encounter problems such as large computational overhead and slow convergence speeds. These difficulties impede real-time application and efficiency in dynamic spectrum environments. Furthermore, the complexity of training DBNs and their susceptibility to parameter adjustment hinder their implementation, resulting in inefficient spectrum sensing and resource allocation in heterogeneous radio frequency environments. The proposed SST-CRN technique efforts to progress the efficiency and accuracy of spectrum sensing, a fundamental task in CRNs, by exploiting CSA's powerful optimization capabilities. The integration with DBNs purposes to take advantage of deep learning's ability to differentiate subtle patterns and correlations in the radio frequency spectrum, which will progress channel identification. Also, this hybrid method determinations to lower the computational complexity and energy consumption connected with classic spectrum sensing approaches, so growing the operational lifespan of CRNs.

This study focuses on the strategy of a novel spectral sensing method for cognitive radio networks (SST-CRN). The presented SST-CRN method contributions in determining the nonlinear threshold based on the chicken swarm process with a deep belief network. The CSA is realistic for optimally choosing the hyperparameters complicated in the DBN perfect. The proposed SST-CRN method includes two phases of processes, namely offline and online. The offline stage creates the nonlinear threshold value to detect energy. Moreover, the online location automatically selects a decision function saved in the offline step to regulate the existence of the main user. A wide range of simulation analyses was performed, and the results are examined under several aspects. Recognizing the crucial role that communication overhead plays in the performance and scalability of optimization algorithms by thoroughly examining communication overhead factors such as message passing, data exchange, and synchronization mechanisms, particularly in the context of large-scale CRNs, this section aims to provide valuable insights into the efficiency and feasibility of using CSA for spectrum optimization. This investigation aims to provide realistic ideas and tactics for addressing communication overhead concerns, hence increasing the applicability and scalability of CSA-based optimization techniques in real-world CRN implementations.

The rest of this essay is organized as follows: The literature study, constraints, and problematic identification for the Enabled SST-CRN were reported in Section 2. Section 3 of this study suggests modeling Chicken Swarm Optimization using Deep Belief Network Enabled Spectrum Sensing Method for Cognitive Radio Networks. The reproduction findings are discussed in Section 4, and conclusion and future study in Section 5.

## 1.1. Contributions

- The SST-CRN model aids in obtaining nonlinear thresholds based on the chicken swarm algorithm (CSA) and a deep belief network (DBN).

- The SST-CRN technique has two operational stages: offline and online. The offline stage generates the nonlinear threshold value for detecting energy.

- Furthermore, the online stage automates the selection of a decision function saved in the offline step for determining the existence of the primary user.

- The performance validation results demonstrated the authority of the projected model and terminated current state-of-the-art methods in standings of various measures.

## 2. Literature review

This section discusses the recently proposed SS approaches in CRN. Wu et al. [10] suggested an IRS-enhanced energy detection for SS. Such circumstances are taken into consideration both with and without a direct connection between the primary and secondary users. The aggregate probability of detection has a closed-form expression thanks to the central limit, and Gamma distributed approximations theorems. To defend from SS data falsification assaults in cognitive radio networks, Wang et al. [11] examine and suggest trust based on collaborative SS data fusion systems. Initially, consider the case in which an integrated information-fusing middle has been established for decision-making. These trust-based data fusion systems rely on the mechanism plan concept to boost success rates and motivate users to provide reliable sensing information.

Mahendru et al. [12] suggested a new arithmetical model for obtaining an optimum sensing duration (sample count) in noise indecision to energy detection technique. The effects of noise improbability on the number of sensed instances have been examined. A new method was proposed for correlating the sensing duration with SNR to obtain desirable performances based on PD and PFA. This model's strengths include simplicity and ease of implementation, as energy detection methods do not need previous knowledge of the main user's signal, making them extremely adaptable to a wide range of signal kinds and settings. This constraint may affect the reliability of spectrum sensing in complex and densely crowded frequency environments.

Awe et al. [13] address the problems of SS in multi-antenna cognitive radio systems with the SVM algorithm. We first formulate the SS problems under several primary user scenarios as several state signal detection problems. On the strength, this method improves detection accuracy greatly by exploiting spatial and temporal information, allowing for more exact identification of accessible spectrum. However, this strategy has weakness. Both beamforming and SVM processing can have significant computational demands, making them difficult to implement in real-time and resource-constrained settings typical of cognitive radio networks.

Ahmad [14] proposed an ML-based detection model to overcome this limit. To tackle the initial constraints, detection is attained by the cyclostationary feature. The regulation of lower SNR, computational cost, and finding detection threshold are tackled by suggesting ensemble classifiers. A dataset is created with distinct orthogonal frequency division multiplexing signals at particular SNR models. Next, the cyclostationary feature is extracted by the FFT accumulation technique. Lastly, the removed part has trained the presented ensemble classifiers for detecting PU signals in lower SNR conditions. Such ensemble classifiers depend on AdaBoost and decision tree algorithms.

Fu et al. [15] explored the soft decision fusion approach for spectrum sensing in CRN. Every SU transmits quantized multi-bit information, which sends local sensing data, rather than original observation statistics/local one-bit complex decision outcomes, to the fusion center. Extensive numerical simulation consequences were used to evaluate the proposed quantization fusion rule in valuation with hard decision, soft fusion, SLMC fusion, and semi-soft fusion rules. Simulation results recognized that the proposed fusion rule accomplished better performance with low computational complexity, and a needed tradeoff between the detection performance and the control channel's communication overhead.

Z. Yongwei et al. [16] came up with a sensing technique based on machine learning as an explanation for the problem of spectrum sensing. This technique relies on the features of the

signal as well as the clustering algorithm that is applied during classification. The k-means clustering procedure defines the variety of the received signals. Calculations of class limits, eigenvalues, and correlations were performed, and the efficacy of the proposed methodology was evaluated. According to the MLA, the likelihood of an error decreased while the detecting display improved. Scalability concerns develop when network size grows; the clustering process may become less efficient, resulting in delays and increased processing time. To ensure that the method is viable in dynamic and heterogeneous wireless environments, a compromise between accuracy and processing efficiency is required.

R. Tallataf et al. [9] predicted a fuzzy logic-based perceptual structure for collaborative energy detection based on a novel link to the principle for local spectrum access. There are three phases to the unclear logic process. The sequencing for blurring, run-in motor, and clearing phases are these. To obtain the most precise forecasts, the nodes' performance is contrasted to the other nodes. The SNR, detection disparities, and threshold are used in these procedures to defend the reliability factor, and the detection performance is evaluated. Its key strength is its capacity to withstand uncertainty and imprecision, which makes it ideal for dynamic and complex radio situations. Fuzzy logic can incorporate a variety of dependability criteria, including as signal intensity, noise levels, and historical detection accuracy, to help make more informed and nuanced decisions concerning spectrum availability. This method's shortcoming in designing and refining fuzzy logic systems is that it can be complex and requires expert knowledge to effectively construct appropriate membership functions and rules.

A. Paul et al. [17] proposed a mathematical model to assess CSS performance in CR-VNs where both PU and SUs move with uniform velocity. Simulation results based on this mathematical analysis display reliable SS even at high velocities of vehicular nodes. Performance gain in $P_d$ values by 2.11% and 6.56% is accomplished correspondingly, at a velocity of 80 km/h. The significance of the number of SUs' decision in the fusion procedure in overall CSS results is also highlighted. Performance gain found in SS reliability at such high speed for the vehicular nodes in metro cities expect to suggestion a important development in DSRC on traffic management and public safety.

New parallel entirely blind multistage detectors have been put out by F. Mashta et al. [18]. The SNR estimator's predicted SNR value determines the appropriate stages. Due to its simplicity and accurate sensing at high SNR, in the initial stage, energy detection is employed. When the SNR is low, extreme eigenvalue detection techniques with different smoothing characteristics are employed at higher settings. The maximum eigenvalue detector's sensing precision rises as the smoothing factor is raised. They also assessed the performance of the two-stage and three-stage variants of the proposed sensor. However, the computational complexity grows as the smoothing factor increases at higher stages, and an eigenvalue detector is used.

It was reported in the research done by A. D. Sahithi et al. [19] regarding the usage of the ED for flexible spectrum access in CR. In this research, the concept of several spectrum sensing technologies as well as their theoretical and mathematical foundations are thoroughly examined. One of the SS approaches that this article examines thoroughly is the ED technique. The underutilized parts of the spectrum are located and made available for further usage using an energy-detecting approach. We can identify and allocate secondary users to spectrum gaps using the energy detection method. On detection performance, fading, shadowing, and buried terminal issues are also covered. Identifying PU signals at low SNR levels is impossible, thanks to the thorough analysis of energy detection methods in this effort.

A. Fawzi et al. [20] forecast two-stage SS techniques for CR using the idea. These methods combine two two-well methods called wavelet carried out to remove and energy detection. In this study, a high SNR value is used to detect whether or not a PU signal exists by associating the energy of the established information to threshold standards using an ED approach. This

was done to ascertain whether or not a PU signal was present. However, before the energy detection (ED) stage, a wavelet denoising stage is employed when a low SNR value is present. When the PU signal is current in noisy environments, this stage is used to detect it and lessen the influence of the noise. The summary of recent investigation paper is listed in Table 1.

The existing methods in CRNs associated to conventional optimization techniques, CSO provides greater convergence rates and precision in high-dimensional search spaces, which creates it ideal for CRNs as of their complex and dynamic structure. This technique, when combined with DBNs, creates use of deep learning's capacity to distinguish complex relationships and patterns in spectrum data, resulting in more accurate and dependable spectrum sensing. In addition to lowering the computational load, this hybrid method progresses the total resource allocation and utilization in CRNs by effectively balancing exploration and exploitation during optimization. Better resilience to non-stationary and heterogeneous radio environments is another feature of the CSO-DBN paradigm that guarantees more stable and interference-resistant communication. Complete, the CSO-DBN method addresses the drawbacks of conventional methods and satisfies the expanding necessities of contemporary wireless networks by offering a more intelligent, flexible, and effective solution for spectrum management.

## 2.1. Limitations

The "Spectrum Sensing Technique for Cognitive Radio Networks" is a promising technique to refining radio spectrum utilization in wireless networks. Though, there are numerous limitations to this technique, which are as follows:

- The accuracy of the spectrum sensing technique is resolute by several factors, including the superiority of the sensing device, the channel characteristics, and interference from other wireless devices. As a result, the technique's accuracy is limited, which can result in false positives and false negatives.

- Spectrum sensing requires time to detect spectrum obtainability in the wireless environment. This significant time delay may impact the operation of the cognitive radio network.

- The performance of the cognitive radio network may be impacted by interference from primary users of the spectrum. This interfering may cause false positives and negatives, resulting in inaccurate spectrum sensing.

- Since cognitive radio networks allow this feature, radio resource devices can use newly nearby spectrums. Though, this requires real-time sensing and decision-making, which can be inspiring to utilization.

## 2.2. Problem identification

A promising explanation for optimizing the use of radio airwaves is increasing: cognitive radio networks. A key constituent of CRNs is spectrum sensing, which permits unauthorized individuals to detect and use available spectrum bands without interrelating with authorized users. On the other together, channel fading, noise insecurity, and errors in signal recognition are still necessary but problematic aspects of the spectrum sensing technique in CRNs that can undermine system performance. One of the major difficulties is the dynamic and impulsive nature of the radio spectrum. Next that of the variable interference levels, signal concentrations, and frequency residence, CRNs find it challenging to use spectrum sensing and decision-making proficiently. Moreover dissimilar has to rapid adapt to changing network

Table 1. A summary of related work in CRN.

| Authors | Year | Techniques | Gaps or concept not covered | Advantages | Disadvantages |
|---|---|---|---|---|---|
| A. Fawzi et al. [21] | 2022 | Two methods ED and WD are combined to propose two-stage spectrum sensing strategies. | The Shannon entropy is the only topic of this paper. It excludes Tsallis, Renyi, and Kapur entropy, among other forms of entropy. | • Easy to implement<br>• Preceding understanding of the main signal properties is not necessary | • Great false alarm rate<br>• Unreliable at low SNR values |
| A. D. Sahithi et al. [19] | 2022 | To address the issue of fading and hidden primary terminal uncertainty, cooperative spectrum sensing with ED is employed. | The suggested approach is not resistant to noise improbability at low SNR, while improving SS performance. Furthermore, there is a significant increase in sensing time and computing complexity since WD and ED are merged at low SNR. | • Robust against noise uncertainty<br>• Differentiate among signal and noise | • Extended sensing duration for optimal performance<br>• High energy consumption when sample sizes are big |
| G. Prieto et al. [21] | 2019 | A detection technique for spectrum sensing based on entropy is suggested. A number of guidelines for figuring out how many bins a histogram has are assessed. Here are those guidelines: Scott rule, Sturges rule, and square root rule | The goal of this work is not to solve the noise uncertainty issue. | • Better detection at low SNR region<br>• Optimal sensing | • Previous knowledge of the main user signal is essential<br>• Impracticable as it's not always possible to have prior knowledge about the signal |
| F. Mashta et al. [18] | 2021 | The discussion of two- and three-stage SS detectors is done in detail.<br>Utilizing various smoothing factors, ED and maximum eigenvalue detector are employed. | Because of its sensitivity to noise uncertainty, SS performs poorly overall at low SNR even with three- and two-stage spectrum sensing. Moreover, the system's entire complexity increases with the employment of the eigenvalue-based detector. | • If properly trained, machine learning can identify can be a useful method.<br>• Minimize the delay of the detection | • Has to be adapted in learning in very fast changing environments<br>• Features assortment affects detection rate and adds complexity |

environments. It is imperative to confirm that the DBN-based sensing technology can promptly adjust to dissimilar in spectral landscape in order to maintain optimal performance. In furthermore, the model's scalability in large-scale CRNs is a significant as the DBN could function as a bottleneck in situations requiring a lot of resources. These complications significant to be fixed if CRNs using DBN-enabled spectrum sensing techniques are to be more reliable and effectively structured.

## 3. The proposed model

An accumulative CSO model for CRNs supported by DBN for spectrum sensing classifies many important challenges. Disabling the computational complexity that results from integrating DBN's deep learning features with CSO is the most difficult element. Finding a happy medium between efficient optimization and quick spectrum sensing is essential, especially for highly dynamic and uncertain radio situations. Modeling becomes more complicated for accomplishing minimal sensing time, which is essential to reacting to changing network conditions in real-time. A thorough comprehension of the interplay among the DBN and CSO components is necessary for the challenging but essential task of coordinating their parameter setting. In order to guarantee the model's scalability in large-scale CRNs as well as resilience and generalization across different contexts, the hybrid technique that has been designated becomes much more intriguing.

Energy detection is an extensively employed model because it has lower complexity and doesn't need previous data regarding the primary signal. With this approach, the decision about spectrum occupancy merely hinges on the noise-based threshold reached [22]. The point determines whether the primary users are available and is related to the observed energy. In essence, it seeks to distinguish among two possible states: prior users signal is empty, signified as $H_0$, or main users signal is current, signified as $H_1$. The following hypotheses have been

tested using the energy detector decision:

$$H_0: \ Y(n) = W(n), : \ \textit{Primary user absent}$$

$$H_1: \ Y(n) = S(n) + W(n), : \ \textit{Primary user present} \tag{1}$$

Where $n = 1,2,3,\ldots,N$ is the example number of the sampled signal that has been received, $Y(n)$ is the sampled signal that secondary users have received, $W(n)$ is the noise presented by the AWGN channel with zero mean and variance $\sigma_n^2$. $s(n)$ is the signal from PU with variance $\sigma_s^2$ and zero mean, and h is the channel's impulse response or the channel amplitude gain between the PU transmitter and secondary user (SU) receiver since we use AWGN channel $h = 1$. $H_0$ and $H_1$ stand for the alternative hypothesis (existence of the PU) and absence (null hypothesis), respectively.

The proposed SST-CRN model's overall operational procedure is shown in Fig 1. At the energy detector. The filtered signal has been later passed by A/D converters. Then, Output of A/D converters are shaped and incorporated, terminating a predetermined period of intermission [23]. The resulting indication is utilized for formulating test statistics. The ED test statistics are shown below:

$$T = \sum_{n=0}^{N} |Y(n)|^2 \tag{2}$$

Where as $n = 0,1,2,3,\ldots,N$, represent the sample count (detection period). Giving to the central limit theorem, the $T$ statistic distributed is Gaussian when $N$ sample numbers are sufficient.

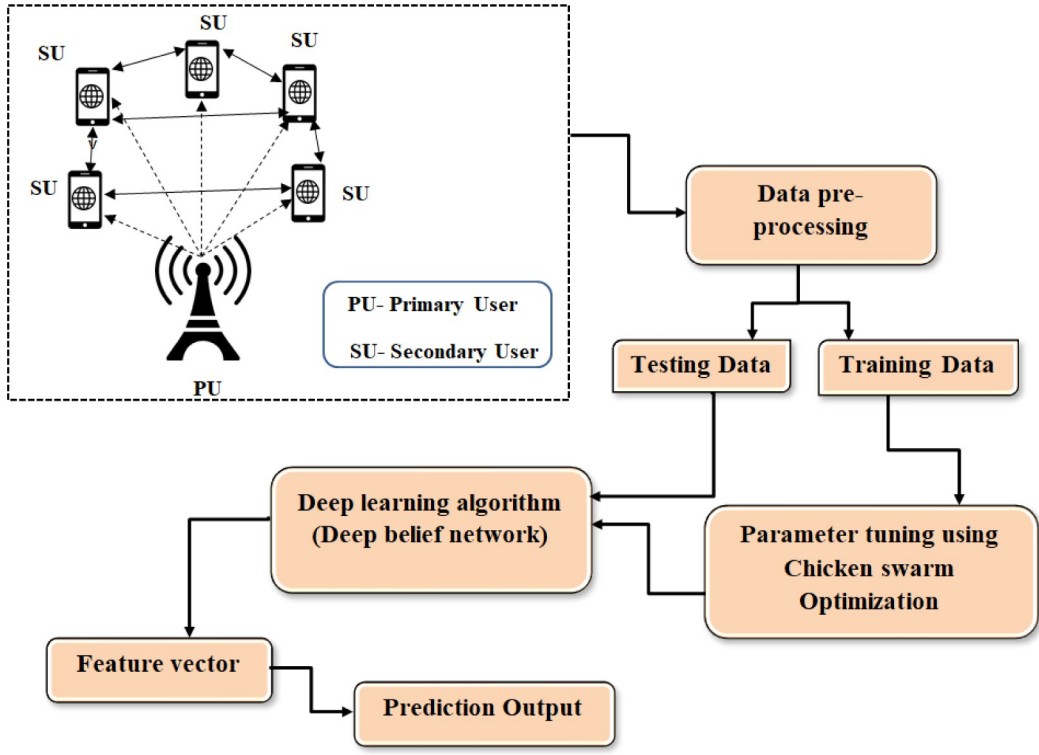

**Fig 1. Illustration for the SST-CRN model.**

The results of the test for binary hypotheses were revised by:

$$H_0: \ T \sim Normal(N\sigma_n^2 + N2\sigma_s^4)$$

$$H_1: \ T \sim Normal((\sigma_n^2 + \sigma_s^2), 2N(\sigma_n^2 + \sigma_s^2)^2) \tag{3}$$

In the equation, $\sigma_s^2 \& \sigma_n^2$ represent the signal modification and noise alteration, correspondingly. Where N is the sample number, T stands for test statistics is related to the inception ($\lambda$) for making the last decision on whether the primary signal has been current. The efficiency of energy detectors is considered three parameters existing according to the test measurements below the binary hypotheses. Allowing the probability of recognition $P_d$, false apprehension $P_{fa}$, and miss finding ($P_m = 1 - P_d$) are represented as

$$P_d = P(T > \lambda/H_1) = Q\left(\frac{\lambda - N(\sigma_n^2 + \sigma_s^2)}{\sqrt{2N(\sigma_n^2 + \sigma_s^2)^2}}\right) \tag{4}$$

$$P_{fa} = P(T > \lambda/H_0) = Q\left(\frac{\lambda - N\sigma_n^2}{\sqrt{2N\sigma_n^4}}\right) \tag{5}$$

In which Q(.) represent an alternative to the Gaussian distribution function. Q-function Q (x) can be formulated by:

$$Q(x) = -\frac{1}{\sqrt{2\pi}}\int_x^\infty \exp\left(-\frac{y^2}{2}\right)dy. \tag{6}$$

The following can be used to determine the detection threshold:

$$\lambda_{ED} = \sigma_\omega^2[\sqrt{2N}Q^{-1}(P_f) + N] \tag{7}$$

## 3.1. Design of CSA-DBN technique

The proposed SST-CRN technique involves the design of a CSA-DBN method for determining the non-linear thresholding values in CRN. Restricted Boltzmann machine (RBM) is a generative stochastic ANN, a different kind of Markov random field that introduces a joint distribution amongst observed information and a hidden feature. It can be two layers of NN which contain visible and remote units. Every visible unit is associated with all hidden units. However, it could not be linked among all the visual and hidden layer nodes. The RBM involves the matrix $W_{ij}$ signifying the weight amongst visible node $v_i$ and hidden node $h_j$, as well as the visible layer bias terms $a_i$ and the hidden layer bias term $b_j$. The weights, as well as biases, have been estimated with maximized marginal distribution over visible units that is determined as:

$$P(v) = \frac{1}{Z}\sum_h e^{-E(v,h;\theta)} \tag{8}$$

Where $\theta = (a_i, b_j, w_{ij})$, and $Z$ refers to the normalization factor that could be estimated as follows:

$$Z = \sum_v \sum_h e^{-E(v,h;\theta)} \tag{9}$$

To provide the model parameter $\theta$, the energy function has been determined as:

$$E(v, h; \theta) = -\sum_{i=1}^{n} \sum_{j=1}^{m} w_{ij} \, v_i h_j - \sum_{i=1}^{n} a_i \, v_i - \sum_{j=1}^{m} b_j \, h_f \qquad (10)$$

The maximal assessment can be attained by estimating the partial logarithmic derivatives in terms of their parameters $a_i, b_j, w_{ij} \in \theta$ utilizing [24]:

$$-\frac{\partial \log(p(v))}{\partial \theta} = -\frac{\mathrm{d}}{d\theta}\left(\log \sum_h \frac{\exp(-E(v,h))}{Z}\right)$$

$$= -\frac{Z}{\sum_h \exp(-E(v,h))}\left(\sum_h \frac{\exp(-E(v,h))}{Z} - \sum_h \frac{\exp(-E(v,h))}{Z^2}\frac{dZ}{d\theta}\right)$$

$$= \sum_h \left(\frac{\exp(-E(v,\mathrm{h}))}{\sum_h \exp(-E(v,\mathrm{h}))}\frac{dE(v,h)}{d\theta}\right) + \frac{1}{Z}\frac{dZ}{\mathrm{d}\theta}$$

$$= \sum_h P(h|v)\frac{dE(v,\mathrm{h})}{d\theta} - \frac{1}{Z}\sum_{v,h}\exp(-E(v,h)) \times \frac{dE(v,h)}{d\theta}$$

$$= \sum_h P(h|v)\frac{dE(v,\mathrm{h})}{d\theta} - \sum_{v,h}P(v,h)\frac{dE(v,h)}{d\theta} \qquad (11)$$

Generally, it is also rewritten as:

$$-\frac{\partial \log(p(v))}{\partial a_i} = \langle v_i \rangle_{data} - \langle v_i \rangle_{\mathrm{model}} \qquad (12)$$

$$-\frac{\partial \log(p(v))}{\partial b_j} = \langle h_j \rangle_{data} - \langle \mathrm{h}_j \rangle_{\mathrm{model}} \qquad (13)$$

$$-\frac{\partial \log(p(v))}{\partial w_{ij}} = \langle v_i h_j \rangle_{data} - \langle v_i h_j \rangle_{\mathrm{model}} \qquad (14)$$

Where the angle brackets have been utilized for denoting the anticipation in the dissemination detailed by the following subscript. The anticipation $\langle v_i h_j \rangle_{\mathrm{model}}$ could not be calculated easily. The expectation is moving toward instances in the model distributions to avoid the maximum computation difficulty. Therefore, it could be evaluated as utilizing the contrastive divergence (CD) technique that is developing a typical manner for training RBM. Fig 2, demonstrates the organization of DBN perfectly [25].

Afterward, the parameters are upgraded as follows:

$$a_i^k = a_i^k + \eta(\langle v_i^k \rangle_{data} - \langle v_i^k \rangle_G)$$

$$= P(v_i = 1|\mathrm{h}) - v_i^k \qquad (15)$$

$$b_j^k = b_j^k + \eta(\langle h_j \rangle_{data} - \langle h_j^k \rangle_G)$$

$$= P(\mathrm{h}_j = 1|v) - P(\mathrm{h}_j = 1|v^k) \qquad (16)$$

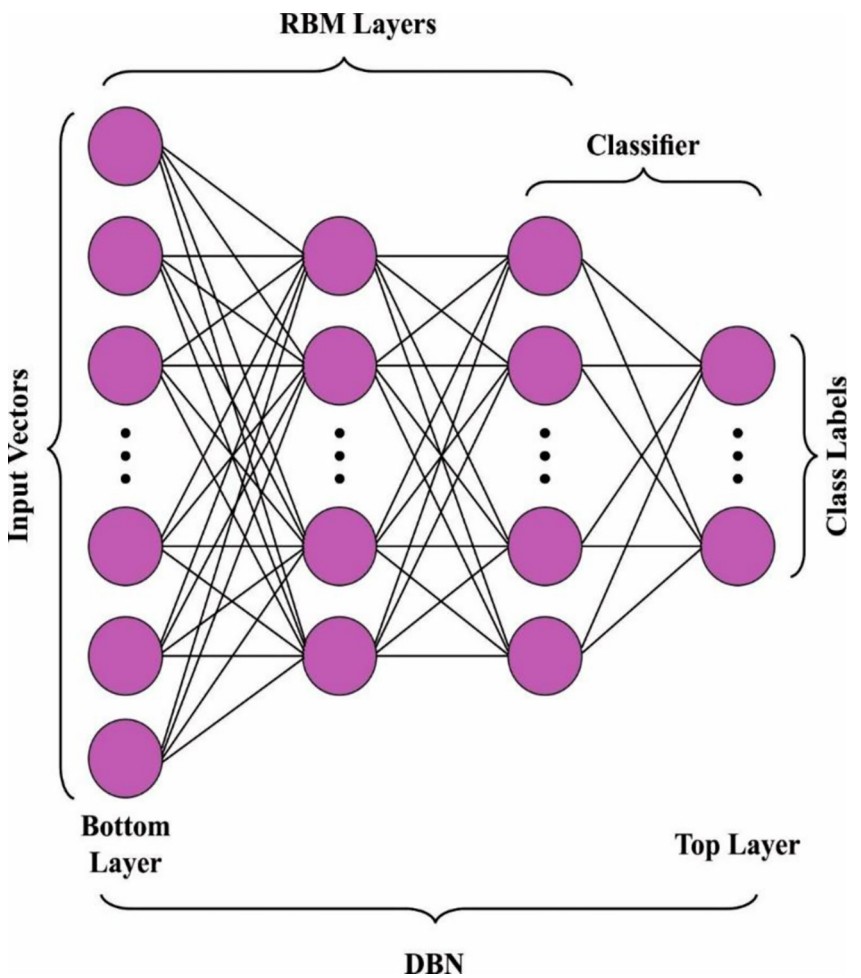

**Fig 2. Structure of DBN model.**

$$w_{ij}^k = w_{ij}^k + \eta(\langle v_i^k h_j \rangle_{data} - \langle v_i^k h_j \rangle_G)$$

$$= P(\mathrm{h}_j = 1|v)v_\mathrm{i} - v_i^k P(h_j = 1|v^k) \qquad (17)$$

Where G implies the Gibbs sampling, $\eta$ refers the rate of learning, and $k$ indicates the $k$th step of CD technique. The conditional probabilities of $v_i$ and $h_j$ are expressed as:

$$P(h_j = 1|v) = g(b_j + \sum_{i=1}^{n} w_{ij}v_i) \qquad (18)$$

$$P(v_i = 1|h) = g(a_i + \sum_{j=1}^{m} w_{ij}h_j) \qquad (19)$$

Where n and m signifies the amount of observable as well as hidden units correspondingly, $g(x) = 1/(1+\exp(-x))$ represents the logistic function utilized as inception.

The CSO algorithm has three different sorts of chicks: hens, roosters, and roles, each with particular behavioral parameters. Provide fundamental assumptions to the CSO algorithm in the following:

1. The CSO algorithm splits chicken swarms into numerous groups, each having a small number of chicks, several hens, and one rooster.

2. The fitness value of chicks, roosters, and hens defines each animal's individuality; the rooster is the fittest animal, while chicks are the most petite fit. Every he chooses one rooster at random as her mate and develops a member of his group, and every chick chooses one hen at random as their mother.

3. Throughout the population, the individual identity, spouse, and mother-children relationships remain unchanged until the G generation (G represents the iterative cycle). The essence, spouse and mother-children relationship would be upgraded at this point.

4. In either particular population, a hen will follow its spouse rooster in search of food and arbitrarily compete with another individual. An individual with the highest fitness value can obtain food.

Every chicken is determined by its location. Assume MN, RN, HN, and CN signify the number of mother hens [26], roosters, hens, and chicks, correspondingly, and $x_{i,j}^t$ represents the position of $i^{th}$ chicken in the $j^{th}$ dimension space on $t^{th}$ iteration, whereas $i \in \{1, \ldots, N\}$, $j \in \{1, D\}$, and $t \in \{1, T\}$ and $N$, $D$, and $T$ correspondingly represents the overall amount of chickens, the dimensional amount, and the maximal iteration time. A chick, rooster, and then have their solid position upgrade formulation. For a rooster, its recurrent position is determined by:

$$x_{i,j}^{t+1} = x_{i,j}^t * (1 + Randn(0, \sigma^2)),\tag{20}$$

$$\sigma^2 = \begin{cases} 1, & \text{if } f_i \leq f_k, \\ \exp\left(\dfrac{f_k - f_i}{|f_i| + \varepsilon}\right), & \text{otherwise } k \in [1, RN], \ k \neq i. \end{cases}\tag{21}$$

Now, $Randn(0, \sigma^2)$ represent an arbitrary assessment following Gaussian dissemination having a probability of zero and a variance of $\sigma^2$, $\varepsilon$ denotes a lesser constant, $k$ indicates the number of additional roosters, i.e., designated subjectively, and $f_i$ and $f_k$ signifies the fitness importance of $i^{th}$ and $k^{th}$ roosters, respectively.

A hen's regular location is determined by the following:

$$x_{i,j}^{t+1} = x_{i,j}^t + C_1 * Rand * (x_{r_1,j}^t - x_{i,j}^t) + C_2 * Rand * (x_{r_2,j}^t - x_{i,j}^t),\tag{22}$$

$$C_1 = \exp\left(\frac{(f_i - f_{r_1})}{(abs(f_i) + \varepsilon)}\right),\tag{23}$$

$$C_2 = \exp(f_{r_2} - f_i).\tag{24}$$

Now, $C_1 \& C_2$ represent the learning factor, $Rand$ signifies an arbitrary assessment following uniform dissemination in the variety of zero and one, $r_1$ signify the index of the rooster i.e., the spouse of $i^{th}$ hen, $r_2$ indicates the amount of a hen/rooster is chosen arbitrarily, and $r_1 \neq r_2$. The recurrent location of a chick is determined by the following:

$$x_{i,j}^{t+1} = x_{i,j}^t + FL * (x_{m,j}^t - x_{i,j}^t),\tag{25}$$

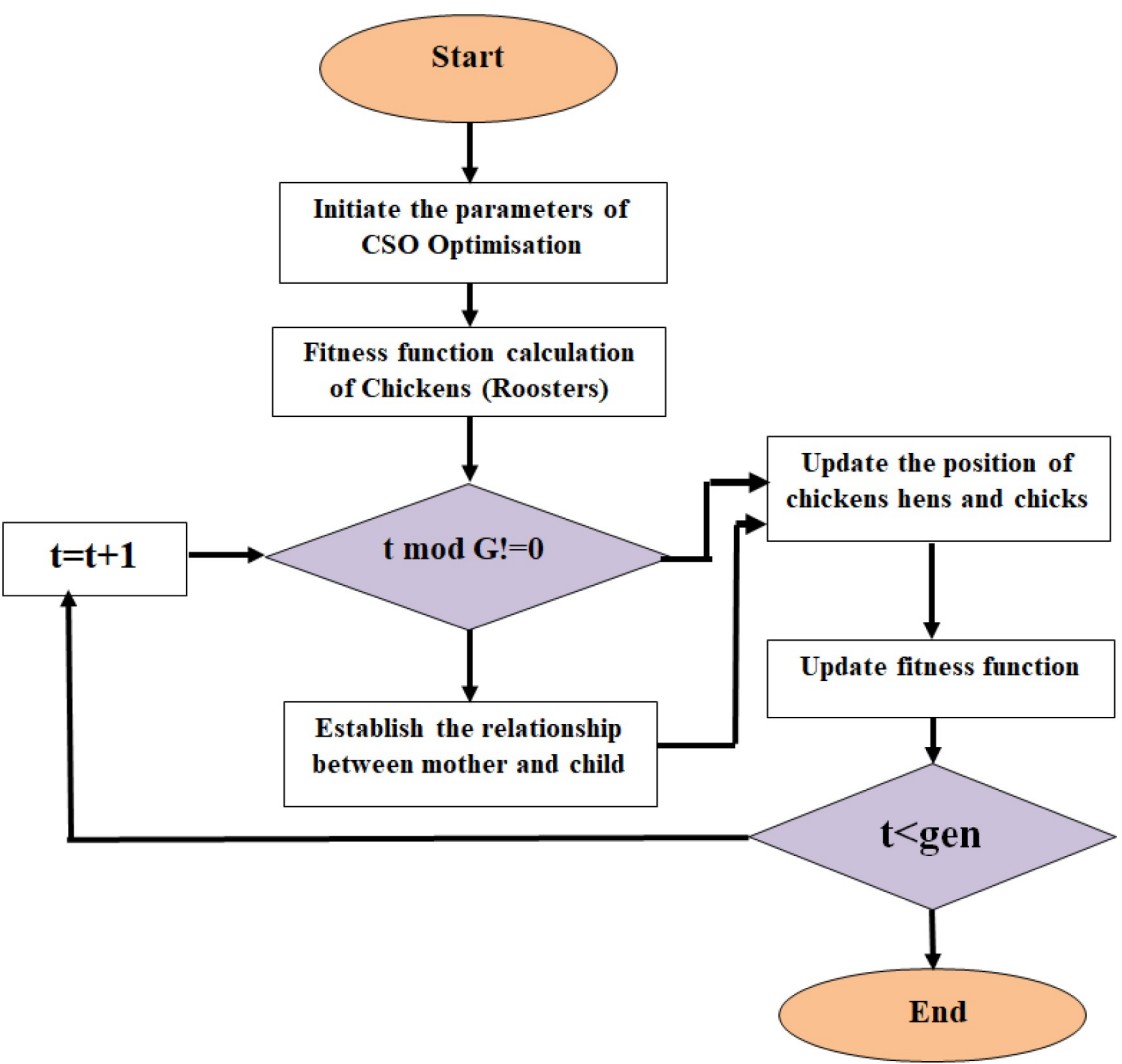

**Fig 3. Flowchart of CSA.**

$x_{m,j}^t$ stands for the mother hen of the chick, while FL stands for random variables between [0, 2]. Hence, the basic CSO algorithm is given in Procedure 1 and Fig 3 shows the flowchart of the CSO algorithm.

```
Algorithm 1: CSO's fundamental algorithm
Input: limitations N,T,RN,HN,CN,MN,G
Output: the optimal explanation
Steps:
(1) Initiate position for chicken arbitrarily.
(2) Compute the fitness values for every chicken, choose the worldwide
optimal location of the populace and the local optimal location of
each chicken, and initiate iteration time t = 1.
(3) When t% G equal to zero (% denotes the remainder operator), sort
each chicken in the descending instruction of fitness value, the opti-
mal RN individual is selected as rooster, the worst CN individual is
chick, and other is hen. Split the population into many groups, each
having chicks, one rooster, and several hens, which arbitrarily select
its spouse and mother, respectively.
```

(4) Upgrade the position for chicks, roosters, and hens based on Eqs (20), (22), and (25), correspondingly, and calculate its fitness value.
(5) Upgrade the global optimal location of the populace and the local optimal location of all individuals.
(6) Iteration time $t = t$+1; when $t$ equal to $M$ or the explanation satisfy accuracy necessities, CSO output the concluding results; otherwise, proceed to Stage 3.

---

Pseudo code of CSO

---

```
1: Prime the populace of chicken as a matrix k;
2: Determines the values of fitness for each row using the k variable;
3: While (t<gen)
4: t = t+1;
5: If (t %G = = 0)
6:  According to their level of physical activity,
k should be split into three groups: the rooster, the hens, and the
chicks.
7: else
8:  For i = 1:
9:   If (i = rooster);
      Upgrade the position of rooster using Eq (20);
    end if
10:   If (i = hen);
      Upgrade the position of hen using Eq (21);
    end if
11:   If (i = chick);
      Upgrade the position of chick using Eq (25);
      end if
12:  Upgrade the new explanation of k
13: end
14: end
```

---

One essential parameter, sensing time, highlights how well the model can sense and adjust to dynamic changes in the radio environment. For fast decision-making and resource allocation, the smallest amount of sensing time must be accomplished. It is important for scalability and practicality to strike a balance among the computing problems of DBN and the difficulties of CSO during training and extrapolation. Also, regularization provisions are important for assuring the generalization of the technique and avoiding overfitting. The robust presentation of the technique through common CRN scenarios depends on finding the ideal balance among exploration and exploitation inside the CSO framework while concerning regularization limitations.

The CSO is an optimization algorithm stimulated by nature, based on the social behaviors and hierarchical structures of swarms of chickens. In CSO, the population is made up of hens, chicks, and roosters, each of which has a thorough function in the optimization procedure. Roosters, illustrative the best explanations, guide the hens and chicks, which designate other candidate descriptions. One type of DL technique called a DBN is made up of countless layers of Restricted Boltzmann Machines (RBMs), which are fundamentally two-layer, basic neural networks. Utilizing the first RBM as the major point, DBNs are trained in an insatiable layer-by-layer fashion, increasingly training new layers while maintaining the stability of the previous trained layers. DBNs are capable of capturing intricate patterns and associates in data thanks to this hierarchical learning method. DBNs are used in the processing and analysis of radio frequency spectrum data in order to classify available channels with high accuracy in spectrum sensing for CRN. By merging CSO with DBN, we can optimize the DBN by using CSO's optimization skills to adjust the DBN's settings and progress spectrum sensing

performance. In dynamic wireless contexts, this synergy guarantees effective spectrum usage and strong available channel detection.

## 3.2. Process involved in offline stage

To overcome the disadvantages mentioned above of standard energy detection, the researchers presented the technique to attain a non-linearly threshold dependent upon SST-CRN. The method procedure encompasses two different modules, namely Offline and Online. An essential function of the Offline Module has to generate non-linear point-to-energy detections. Initially, the presented technique makes a trained signal and trained noise in AS1-3. So, the difference in conditioned signal has been identified value $\sigma^2_{s_{tr}} = \lambda_{tr}$, the difference in trained noise is $\sigma^2_{u_{tr}}, = 1$, and trained $SNR$ is $\lambda_{tr}$. Secondly, according to the parameter $\lambda_{tr}$ and amount of signal instances $N_s$, 2 classes of trained decision statistics in hypothesis $H_0$ and hypothesis $H_1$ has been attained that are provided as:

$$T_{1_{tr}}^{N_s,\lambda_{tr}} = \frac{1}{N_s} \sum_{n=1}^{\lambda_s} |s_{tr}(n) + u_{tr}(n)|^2 \tag{26}$$

$$T_{0_{tr}}^{N_s} = \frac{1}{N_s} \sum_{n=1}^{N_s} |u_{tr}(n)|^2 \tag{27}$$

Where the group of non-central chi-square variables with a mean of $1+\lambda_{tr}$, $2N_s$ degrees of freedom [27], and non-centrality parameters $\lambda_{tr}$ is denoted by the symbol $T_{1_{tr}}^{N_s,\lambda_{tr}}$. The core chi-square variable class, however, is represented by the symbol $T_{0_{tr}}^{N_s}$, which has a mean of 1 and $2N_s$ degrees of freedom.

Next, the labelling of all variables of $T_{1_{tr}}^{N_s,\lambda_{tr}}$ class as "+1", and all variables of $T_{0_{tr}}^{N_s}$ period as "−1", afterward, these 2 periods of variables have been utilized as trained information for training CSO-DBN. Therefore, the separating hyperplane $\langle w^* \cdot x \rangle + b = 0$ and decision functions $f(x) = sign(\langle w^* \cdot x \rangle + b)$ has been resultant. Finally, the variables of $T_{0_{tr}}^{N_s}$ class are implemented for testing this decision functions given that for gaining the probabilities (referred as $P_e^{N_s,\lambda_{tr}}$) which the decision function incorrectly label a $T_{0_{tr}}^{N_s}$' variable as $T_{1_t}^{N_s,\lambda_{tr}}$ variable.

## 3.3. Process involved in online stage

During the Online module, the presented technique automatically selects most of the decision functions (based on need the number of signal instances $N_s$ and probabilities of false alarm $P_f$) saved from the Offline Module as nonlinear inceptions for judging if the actual primary users exist, e.g., the needed amount of signal instances have been $N_s = 5$ and probabilities of false alarm are $P_f = 0.1$, the presented technique is implemented decision functions $f_{5,01}(x)$ as the non-linear threshold. When the decision functions $f_{N_s,P_f}(x)$ has been selected as non-linear threshold as:

$$P_f = \Pr(f_{N_s,P_f}(T\prime) = 1 | H_0) = P_e^{N_s,\lambda_{tr}} \tag{28}$$

$$P_d = \Pr(f_{N_s,P_f}(T\prime) = 1 | H_1) \tag{29}$$

The outcome of SS has been provided as follows:

$$\begin{cases} f_{N_s,P_f}, (T\prime) = 1, \text{Primary user has been projected (hypothesis} H_1) \\ f_{N_s,P_f}, (T\prime) = -1, \text{Primary user has been projected (hypothesis } H_0) \end{cases} \tag{30}$$

# 4. Result and discussion

The simulation results for energy detection, two-stage detection using MATLAB version 2020a, and entropy-based detection are covered in this section. The efficiency of the suggested two-stage SS is assessed through simulations in comparison to conventional and entropy-based energy detection in AWGN channels. The simulation results were generated using 10,000 Monte Carlo runs and 1,000 example numbers. The IEEE 802.22 standard states that any cognitive radio simulation shall account for the necessary detection probability ($\geq$90%), the risk of a miss-detection ($<$10%), and the possibility of a false alarm ($\leq$ 10%). One important factor that is evaluated in each of the result graphs is the SNR wall, which is compared between the proposed methods and the existing ones. Furthermore, the SNR wall is employed for comparative analysis of various entropy-based detection techniques. The lowest SNR below which detection is impossible is known as the SNR wall. In the circumstance of a cognitive radio network, a Remote Monitoring Logging System Service in Cognitive Radio Network (RMLSSCRN) frequently entails a system for remotely monitoring and logging tasks associated with the upkeep of the cognitive radio infrastructure.

## 4.1. Parameter setting

To whole optimal performance, it is significant to set parameters for CSO in CRN using Deep Belief Network (DBN)-enabled spectrum sensing. A population size of 20–50 is frequently designated to permit for a thorough search space exploration while maintaining reasonable processing requirements. Another significant parameter is the maximum number of iterations, which is often set between 100 and 500 iterations to provide the algorithm sufficient time to converge to an ideal clarification without generating undue delay a significant factor in contexts with dynamic spectrum. Similarly, CSO-specific characteristics, with the quantity of leaders and followers, are set up to preserve a balance between the search space's inspection and utilization. The complexity of the CRN environment and the available computational resources are taken into consideration when fine-tuning these settings, which are tested empirically. The meticulous adjustment of parameters guarantees that the CSO algorithm combines with the DBN in an efficient and accurate manner for spectrum sensing, hence improving the whole performance of the network. All of the comparative tests in this study are conducted using Python 3.8.5 in a PC with 8GB of RAM [28]. Let us assume that the population size is 50, the maximum number of iterations is 500, the α in the fitness function is set to 0.9999, and there are 20 independent running experiments on the datasets. The selection scheme corresponding to each particle is tested for classification accuracy using the classifier.

The benchmark approaches RMLSSCRN-100 and RMLSSCRN-300 possess certain important features that improve their efficiency in spectrum sensing in CRNs. Both approaches apply Random Matrix Learning-based Spectrum Sensing (RMLSS) methods, which accomplishment principles from random matrix theory and machine learning algorithms. These approaches permit the change amongst noise and signal components in the spectrum, which is vital for accurate spectrum sensing. In totalling, RMLSSCRN-100 and RMLSSCRN-300 use sophisticated algorithms and particular feature sets to development the accuracy and dependability of spectrum detection. These algorithms probably include preprocessing enhancements to growth raw input data, feature selection methods to determine important spectral properties, and model changes to advancement entire performance. Likewise, these methods may use statistical features of received signals, such as eigenvalue decomposition and hypothesis testing, to aid in generating spectrum sensing decisions. In whole, RMLSSCRN-100 and RMLSSCRN-300 exhibit advanced features intended to progress spectrum sensing in CRNs, creating them valuable values for concerning with other tactics.

At great SNR levels, the main user's signal is challengingly stronger than the noise, subsequent in greater detection probability and lower false alarm rates due to the clear parting amongst signal and noise. In difference, at low SNR stages, the signal is closer in magnitude to the noise, generating it harder to differentiate among the two. This leads to reduced detection probabilities and superior false alarm rates as the sensing algorithm fails to resourcefully categorize the presence of the major user between the noise. The performance of spectrum sensing methods is intensely dependent on SNR levels, with inordinate SNR following in superior detection probability and low SNR providing considerable difficulties. Other significant elements that influence detection performance include sensing time, channel conditions, noise characteristics, and signal qualities. Optimizing these aspects using modern algorithms and adaptive sensing approaches can considerably development the reliability and effectiveness of spectrum sensing in cognitive radio networks. In practically, it leads to more efficient, reliable, and adaptable spectrum sensing in cognitive radio networks, which progresses overall communication quality and device performance. Theoretically, it stimulates the improvement of sophisticated algorithms and models that push the limits of present spectrum sensing capabilities, driving improvement and development in wireless communications.

This unit perfectly examines the SST-CRN's presentation with existing methods in terms of dissimilar evaluation measures. Initially, the probability of detection (Pd) results in the analysis of the SST-CRN model takes place under non-fading channels with varying SNR values in Table 2 and Fig 4, The results depicted that the SST-CRN model has obtained an effective outcome with the maximum Pd under every SNR. For instance, with SNR of -24dB, the SST-CRN perfect has increased a developed Pd of 0.810, whereas the RMLSSCRN-100 and RMLSSCRN-300 methods have accomplished a lower Pd of 0.577 and 0.736, respectively. Also, with SNR of -22dB, the SST-CRN perfect has gained a developed Pd of 0.830, whereas the RMLSSCRN-100 and RMLSSCRN-300 methods have accomplished a lower Pd of 0.597 and 0.793, respectively.

**Probability of detection.**　The possibility that a cognitive radio system can accurately detect the existence of a primary user signal through background noise or interference,

**Table 2. Probability of detection analysis of SST-CRN model under non-fading channels.**

| Pd (Non-Fading Channel) | | | |
|---|---|---|---|
| SNR | RML SSCRN-100 | RML SSCRN-300 | SST-CRN |
| -24 | 0.577 | 0.736 | 0.801 |
| -23 | 0.582 | 0.757 | 0.812 |
| -22 | 0.597 | 0.793 | 0.830 |
| -21 | 0.597 | 0.800 | 0.839 |
| -19 | 0.604 | 0.831 | 0.869 |
| -18 | 0.613 | 0.842 | 0.885 |
| -17 | 0.624 | 0.869 | 0.905 |
| -16 | 0.648 | 0.931 | 0.961 |
| -15 | 0.671 | 0.961 | 0.982 |
| -14 | 0.693 | 0.967 | 0.973 |
| -13 | 0.825 | 0.973 | 0.976 |
| -12 | 0.925 | 0.979 | 0.982 |
| -11 | 0.983 | 0.989 | 0.992 |
| -10 | 0.993 | 0.995 | 0.997 |

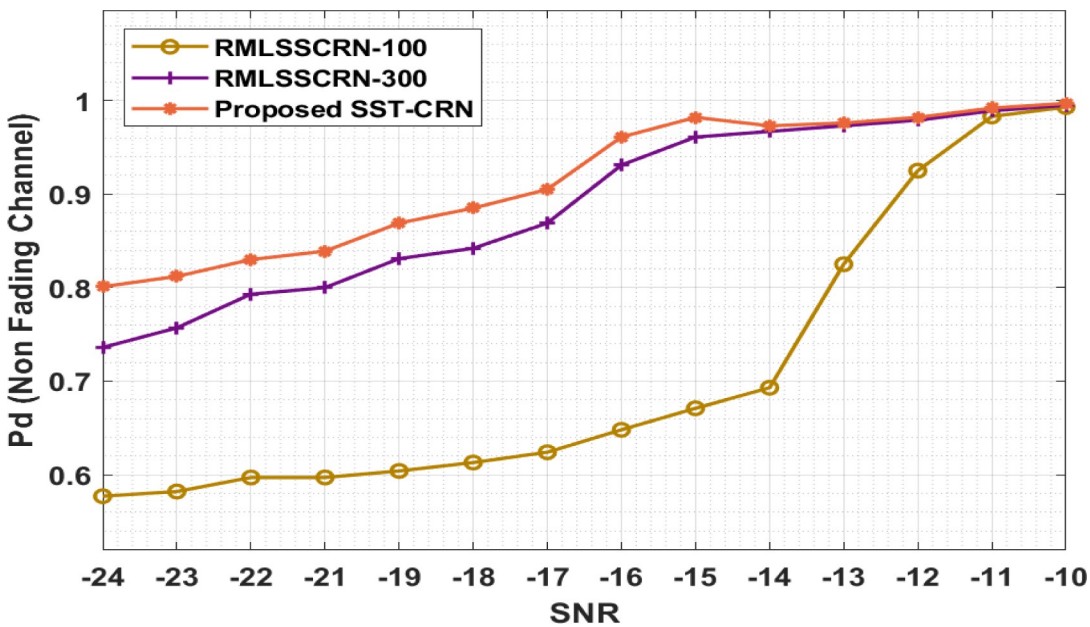

**Fig 4. Pd analysis of the SST-CRN model under non-fading channels.**

permitting effective use of available spectrum resources, is known as the probability of detection.

$$P_d = Q\left(\sqrt{\frac{2\gamma}{N_0}}\right) \tag{31}$$

Where:

$Q(x)$ standard normal distribution's complementary cumulative distribution function is denoted as Q(x).

$\gamma$ is the SNR of the received signal.

The noise power spectral density is denoted by $N = 0$.

**Pnf.** Probability non-fading channels refers to the possibility of meeting wireless communication channels that do not exhibit considerable fading over time during the spectrum sensing process of a cognitive radio network. This method entails finding and utilizing available spectrum opportunities in stable channels in order to progress the efficiency and reliability of spectrum usage in cognitive radio networks.

Table 3 and Fig 5 display the SST-CRN model analysis results regarding spectral hole exploitation probability (Pnf). The findings show that, across a range of SNR values, the SST-CRN model has accomplished maximum performance with larger Pnf. For instance, with -SNR of -24dB, the SST-CRN model has increased Pnf of 0.761, whereas the RMLSSCRN-100 and RMLSSCRN-300 methods have provided a condensed Pnf of 0.539 and 0.694. Also, with a -14dB SNR, the SST-CRN model yields a developed Pnf of 0.940, while the RMLSSCRN-100 and RMLSSCRN-300 approaches yield lower Pnf values of 0.864 and 0.862. In conclusion, the SST-CRN model yielded an enhanced Pnf of 0.996 with a -SNR of -10dB, whereas the RMLSSCRN-100 and RMLSSCRN-300 methods produced a reduced Pnf of 0.988 and 0.993.

Table 4 and Fig 6 present a probability of error analysis of the SST-CRN approach with existing ones. The results showed improved performance with the minimal Pe. To illustrate, the SST-CRN model has a lower Pnf of 0.0145 with a -SNR of -24dB, whereas the

**Table 3. Analysis of SST-CRN model results for spectral hole exploitation probability.**

| Pd (Non-Fading Channel) | | | |
|---|---|---|---|
| SNR | RML SSCRN-100 | RML SSCRN-300 | SST-CRN |
| -24 | 0.539 | 0.694 | 0.761 |
| -23 | 0.607 | 0.691 | 0.805 |
| -22 | 0.634 | 0.688 | 0.805 |
| -21 | 0.658 | 0.691 | 0.808 |
| -19 | 0.677 | 0.699 | 0.824 |
| -18 | 0.688 | 0.696 | 0.829 |
| -17 | 0.707 | 0.729 | 0.843 |
| -16 | 0.729 | 0.734 | 0.840 |
| -15 | 0.789 | 0.802 | 0.900 |
| -14 | 0.864 | 0.862 | 0.940 |
| -13 | 0.927 | 0.930 | 0.989 |
| -12 | 0.978 | 0.984 | 0.992 |
| -11 | 0.984 | 0.984 | 0.989 |
| -10 | 0.988 | 0.993 | 0.996 |

RMLSSCRN-100 and RMLSSCRN-300 approaches have produced higher Pnf values of 0.0536 and 0.145. In the meantime, the SST-CRN technique has produced an increased Pnf of 0.0118 with a -SNR of -19dB, whilst the RMLSSCRN-100 and RMLSSCRN-300 models have produced a decreased Pnf of 0.0284 and 0.0244. Followed by, with -SNR of -14dB, the SST-CRN technique has resulted in an increased Pnf of 0.0019 while the RMLSSCRN-100 and RMLSSCRN-300 models have given a reduced Pnf of 0.0045 and 0.0045. Eventually, with -SNR of -10dB, the SST-CRN method has increased Pnf of 0.0000, while the RMLSSCRN-100 and RMLSSCRN-300 models have given a reduced Pnf of 0.0011 and 0.0006.

At first, the probability of detection (Pd) results from analyses of the SST-CRN technique takes place under fading channels with varying SNR values in Table 5 and Fig 7, The result

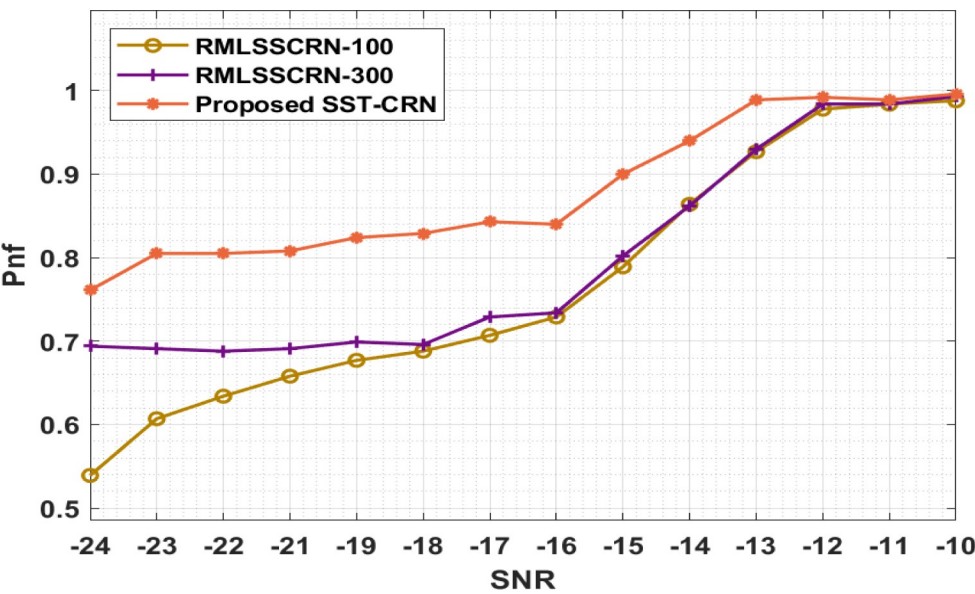

**Fig 5. Pnf analysis of the SST-CRN model.**

**Table 4. Probability of error analysis of SST-CRN model under non-fading channels.**

| Pe (Non-Fading Channel) | | | |
|---|---|---|---|
| SNR | RMLSS CRN-100 | RML SSCRN-300 | SST-CRN |
| -24 | 0.0536 | 0.0244 | 0.0145 |
| -23 | 0.0410 | 0.0244 | 0.0152 |
| -22 | 0.0351 | 0.0244 | 0.0138 |
| -21 | 0.0291 | 0.0238 | 0.0132 |
| -19 | 0.0284 | 0.0244 | 0.0118 |
| -18 | 0.0271 | 0.0231 | 0.0085 |
| -17 | 0.0211 | 0.0178 | 0.0092 |
| -16 | 0.0178 | 0.0165 | 0.0079 |
| -15 | 0.0118 | 0.0105 | 0.0045 |
| -14 | 0.0045 | 0.0045 | 0.0019 |
| -13 | 0.0019 | 0.0010 | 0.0007 |
| -12 | 0.0015 | 0.0008 | 0.0003 |
| -11 | 0.0012 | 0.0007 | 0.0000 |
| -10 | 0.0011 | 0.0006 | 0.0000 |

shows that the SST-CRN method has attained remarkable result with the maximum Pd under every SNR. For instance, with SNR of -23dB, the SST-CRN perfect has gained a higher Pd of 0.599, whereas the RMLSSCRN-100 and RMLSSCRN-300 methods have accomplished a lower Pd of 0.497 and 0.510, respectively. In addition, with SNR of -19dB, the SST-CRN process has attained a higher Pd of 0.644, while the RMLSSCRN-100 and RMLSSCRN-300 methods have obtained a lower Pd of 0.531 and 0.561, correspondingly. Furthermore, with SNR of -10dB, the SST-CRN method has attained a higher Pd of 0.874, while the RMLSSCRN-100 and RMLSSCRN-300 approaches have obtained a lower Pd of 0.679 and 0.810, correspondingly.

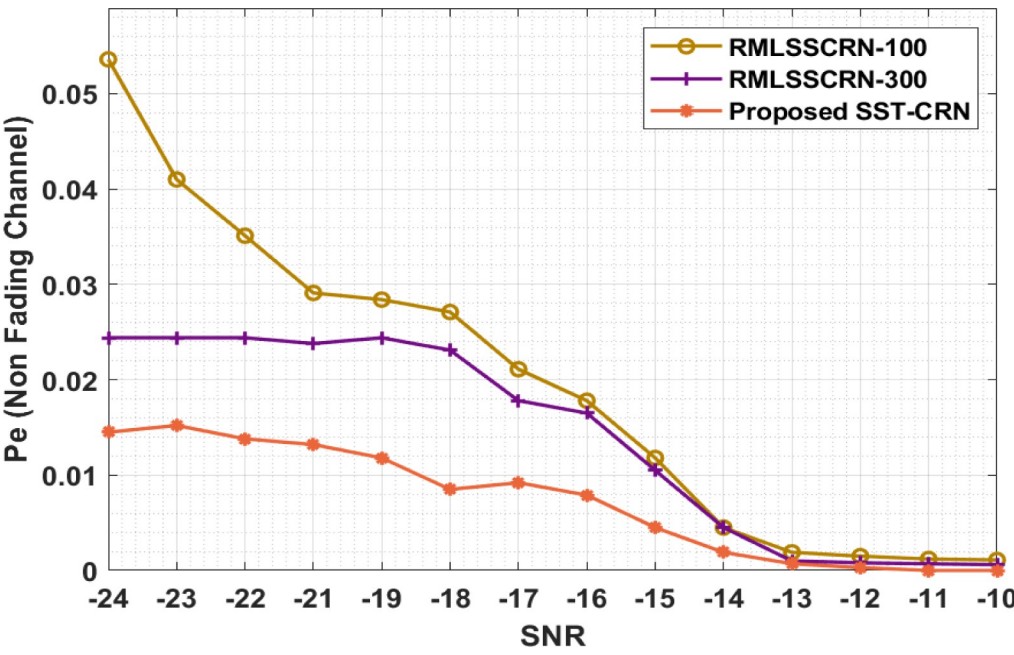

**Fig 6. Pe analysis of the SST-CRN model under non-fading channels.**

**Table 5. Probability of detection analysis of SST-CRN model under fading channels.**

| Pd (Fading Channel) | | | |
|---|---|---|---|
| SNR | RMLSS CRN-100 | RML SSCRN-300 | SST-CRN |
| -23 | 0.497 | 0.510 | 0.599 |
| -21 | 0.493 | 0.513 | 0.601 |
| -19 | 0.531 | 0.561 | 0.644 |
| -17 | 0.570 | 0.663 | 0.728 |
| -14 | 0.592 | 0.774 | 0.823 |
| -12 | 0.637 | 0.774 | 0.860 |
| -10 | 0.679 | 0.810 | 0.874 |

Table 6 and Fig 8, show the result analyses of the SST-CRN method in terms of Pmd. The result demonstrates that the SST-CRN technique has attained higher performances with higher Pmd under varying SNR values. E.g., with a CR user of 2, the SST-CRN method has resulted in an increased Pmd of 0.653, while the RMLSSCRN-100 and RMLSSCRN-300 methods have given a reduced Pmd of 0.552 and 0.575. Also, with a CR user of 10, the SST-CRN process has increased Pmd by 0.752, while the RMLSSCRN-100 and RMLSSCRN-300 methods have a reduced Pmd of 0.576 and 0.706.

**Pmd.** The probability of miss alarm (Pmd) in Cognitive Radio Networks Enabled SST depicts the possibility of failing to identify the attendance of a primary user signal when it is present, indicating a potential inefficiency in spectrum sensing in cognitive radio systems.

$$P_{miss} = 1 - \prod_{i=1}^{N} P_{\text{detection}_i} \qquad (32)$$

Where:

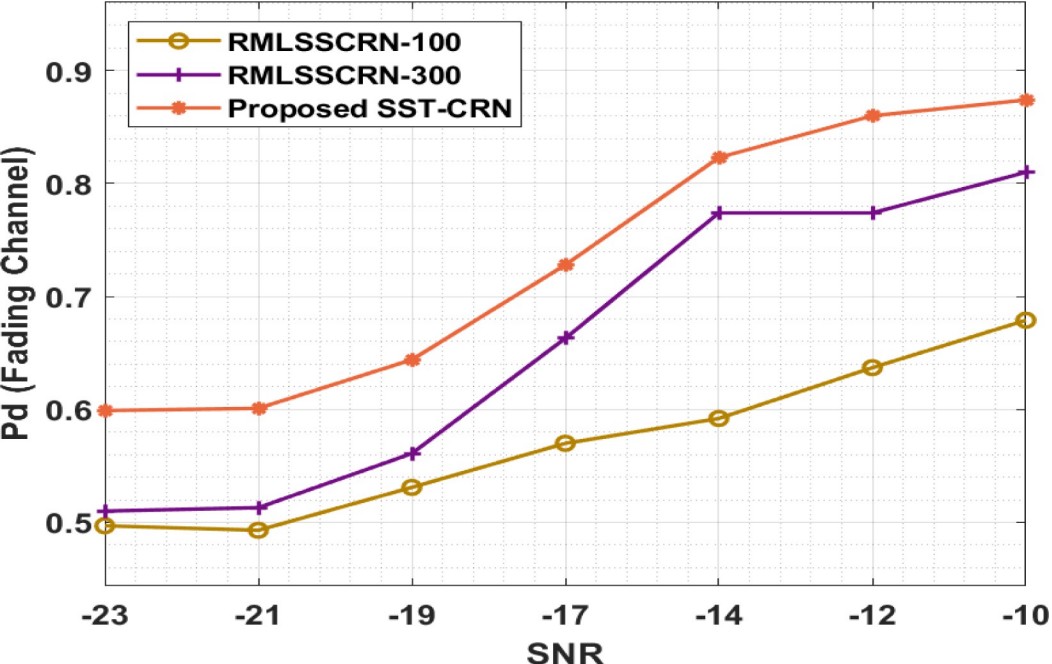

**Fig 7. Pd analysis of the SST-CRN model under fading channels.**

**Table 6. Pmd analysis of the SST-CRN model with several CR users.**

| | Pmd | | |
|---|---|---|---|
| Number of CR users | RML SSCRN-100 | RML SSCRN-300 | SST-CRN |
| 2 | 0.552 | 0.575 | 0.653 |
| 4 | 0.562 | 0.616 | 0.671 |
| 6 | 0.562 | 0.636 | 0.682 |
| 8 | 0.566 | 0.640 | 0.714 |
| 10 | 0.576 | 0.706 | 0.752 |
| 12 | 0.600 | 0.716 | 0.762 |
| 14 | 0.622 | 0.738 | 0.778 |
| 16 | 0.625 | 0.749 | 0.808 |
| 18 | 0.649 | 0.781 | 0.826 |
| 20 | 0.696 | 0.794 | 0.833 |

$P_{miss}$ the likelihood of missing an alert, or going unreported, is denoted as P (miss).

N represents the total number of CR users in the network.

$P_{detection_i}$ Detection I is the likelihood of detection for the i-th CR user.

The probability of error analyses of the SST-CRN techniques with the present one occurs in Table 7 and Fig 9, The result demonstrates the remarkable performances with the minimal Pe. E.g., with -SNR of -24dB, the SST-CRN method has resulted in a reduced Pe of 0.066 while the RMLSSCRN-100 and RMLSSCRN-300 models have given a higher Pe of 0.241 and 0.098. Simultaneously, with -SNR of -19dB, the SST-CRN method has an increased Pe of 0.065 while the RMLSSCRN-100 and RMLSSCRN-300 approaches have a reduced Pe of 0.084 and 0.084. Then, with -SNR of -14dB, the SST-CRN system has an increased Pe of 0.057, while the

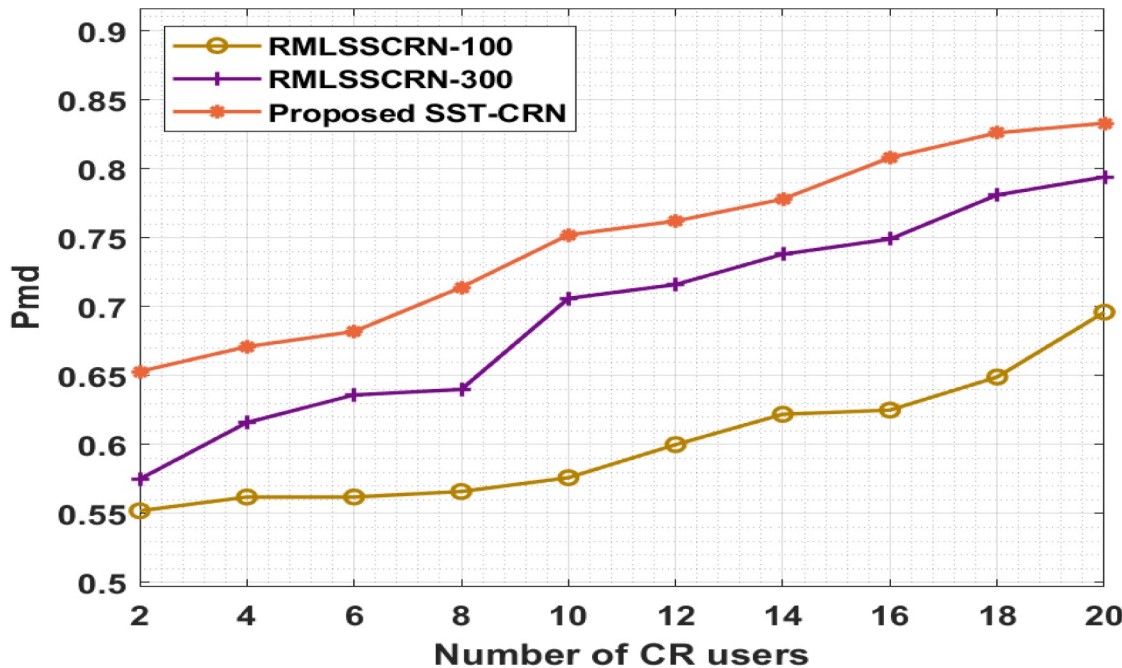

**Fig 8. Pmd analysis of the SST-CRN model.**

**Table 7. Probability of error analysis of SST-CRN under fading channels.**

| Pe (Fading Channel) | | | |
|---|---|---|---|
| SNR | RML SSCRN-100 | RML SSCRN-300 | SST-CRN |
| -24 | 0.241 | 0.098 | 0.066 |
| -23 | 0.108 | 0.084 | 0.067 |
| -21 | 0.088 | 0.084 | 0.064 |
| -20 | 0.083 | 0.084 | 0.064 |
| -19 | 0.084 | 0.084 | 0.065 |
| -17 | 0.086 | 0.084 | 0.066 |
| -16 | 0.085 | 0.085 | 0.062 |
| -14 | 0.085 | 0.077 | 0.057 |
| -13 | 0.086 | 0.071 | 0.056 |
| -11 | 0.083 | 0.063 | 0.054 |
| -10 | 0.084 | 0.051 | 0.050 |

RMLSSCRN-100 and RMLSSCRN-300 methods have a reduced Pe of 0.085 and 0.077. Finally, with -SNR of -10dB, the SST-CRN approach has resulted in an increased Pe of 0.050, while the RMLSSCRN-100 and RMLSSCRN-300 systems have given a reduced Pe of 0.084 and 0.051.

**Probability of error.** The probability of error detection is the possibility that this technique will correctly identify communication-ready spectrum bands while reducing the possibility of missing or mis detecting signals, which is essential for effective spectrum use and dependable network performance.

$$P_e = 1 - \prod_{i=1}^{N}(1 - P_{e_i}) \qquad (33)$$

$P_e$ is the whole probability of error in the SST-CRN method with multiple CR users.
$P_{e_i}$ is the likelihood of error for the i-th CR user.

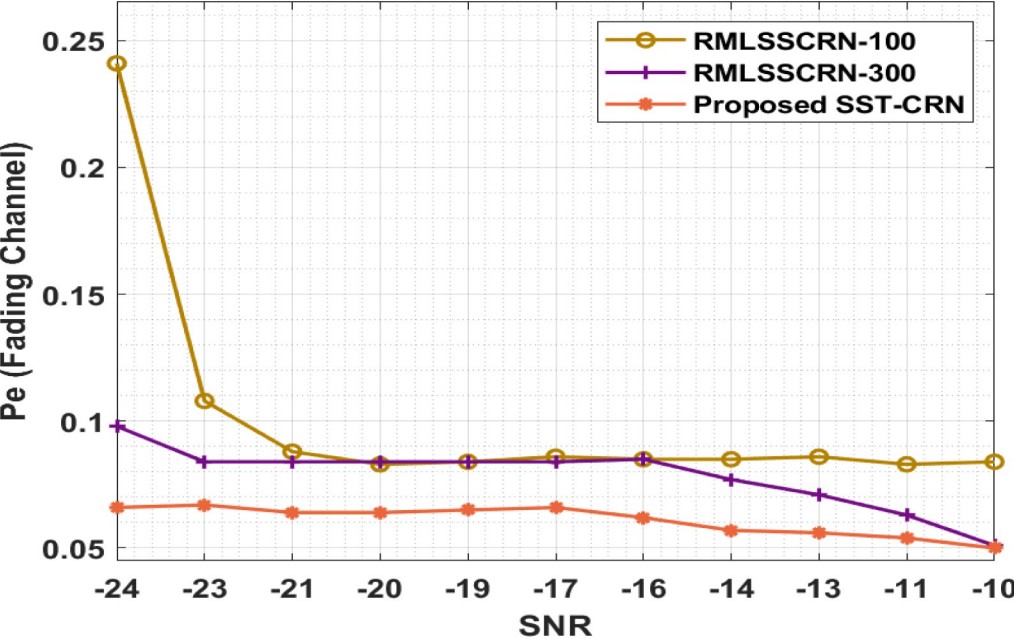

**Fig 9. Pe investigation of the SST-CRN under fading channels.**

## 4.2. Limitations

The proposed SST-CRN approaches display possible for developing spectrum sensing in CRNs, but it also faces possible limitations. First challenge is the computational difficulty of training and deploying DL models such as DBNs, which may limit real-time implementation, mostly in resource-constrained CRN devices. Furthermore, the interpretability of DBN-based spectrum sensing judgments may be problematic, making it difficult to explain and confirm the reasons behind certain detection results. Lastly, while DBNs excel at learning complex patterns, they may struggle in contexts with fast altering dynamics or adversarial conditions, necessitating constant adaptation and refining to remain effective. Addressing these restrictions is critical to realizing the suggested model's full potential in real-world CRN applications.

## 5. Conclusion

Spectrum sensing is a vital module of cognitive radio and a vital technique for enhancing spectrum usage and attaining a rational and orderly distribution of spectrum resources, both of which have considerable research implications and probable applications. In order to handle the growing number of concerns in this field, researchers are assimilating novel techniques like deep learning and deep reinforcement learning, even if classic spectrum-sensing algorithms are no longer appropriate for the more complex detecting environment. In this study presents a refining the performance of CRN the incorporation of CSO, a bio-inspired optimization algorithm stimulated by the combined behavior of chickens, with DBN-enabled spectrum sensing, this method offers a novel solution for addressing the spectrum scarcity problem and improving spectrum utilization competence in CRNs. By leveraging CSO, the proposed model enhances the organization of cognitive radios and the distribution of spectrum resources, allowing CRNs to adapt dynamically to altering environmental circumstances and user difficulties. Also, the integration of DBN-based spectrum sensing enhances the cognitive competences of CRNs by enabling accurate and adaptive spectrum sensing in complex and dynamic radio environments. The major limitation of the proposed method SST-CRN is the high computational complexity and resource requirements of training and deploying DBNs, which can be difficult in situations when resources are limited or real-time. Furthermore, despite its effectiveness in optimization, the CSO algorithm can occasionally experience problems with convergence speed and become stuck in local optima, particularly in contexts with a highly dynamic and heterogeneous spectrum. The synergistic combination of CSO modeling and DBN-enabled spectrum sensing establishes important probable for enhancing the reliability, efficiency, and scalability of CRNs, eventually enabling the realization of more intelligent and autonomous wireless communication systems. The SST-CRN technique aids in the finding of nonlinear thresholds based on the CSA-DBN model, in which the DBN model's parameters are ideally designated using the CSA. A wide range of simulation analyses were accepted, with the results examined in several ways. When DBN's deep learning capabilities are paired with CSO's natural nature-inspired algorithms, a synergistic framework is formed that allows CRNs to explore and allocate frequencies independently with astounding accuracy. The factors such as likelihood of detection, SNR of -24dB, and the SST-CRN perfect have enhanced the developed probability of detection of 0.810, whereas the RMLSSCRN-100 and RMLSSCRN-300 methods have achieved a lesser probability of detection of 0.577 and 0.736, respectively. The parameters such as probability of error analysis with SNR of -24dB, the SST-CRN technique has resulted in a lower probability of error of 0.066 when compared to the existing model

In Future research could focus on creating more efficient and scalable versions of CSO in order to improve its convergence features and reduce computational overhead. To address

local optima issues, hybrid approaches combining CSO with other optimization techniques may be investigated. Furthermore, advances in lightweight and real-time deep learning models may make DBN-enabled spectrum sensing more useful for CRNs. Adaptive learning techniques could also be investigated, allowing DBNs to continuously learn and adapt to different spectrum settings with less input. Finally, incorporating modern data fusion algorithms to collect and process a variety of spectrum sensing data sources could increase the technique's dependability and accuracy.

## Author Contributions

**Conceptualization:** Saraswathi M.

**Data curation:** Saraswathi M., Logashanmugam E.

**Formal analysis:** Saraswathi M., Logashanmugam E.

**Investigation:** Saraswathi M., Logashanmugam E.

**Methodology:** Saraswathi M.

**Project administration:** Saraswathi M., Logashanmugam E.

**Resources:** Saraswathi M.

**Software:** Saraswathi M.

**Supervision:** Saraswathi M., Logashanmugam E.

**Validation:** Saraswathi M.

**Visualization:** Saraswathi M., Logashanmugam E.

**Writing – original draft:** Saraswathi M.

**Writing – review & editing:** Saraswathi M., Logashanmugam E.

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
