## [Decision Letter · Decision Letter 0]

26 Dec 2023

PONE-D-23-25088Chicken Swarm Optimization Modelling for Cognitive Radio Networks Using Deep Belief Network-Enabled Spectrum Sensing TechniquePLOS ONE

Dear Dr. M,

Thank you for submitting your manuscript to PLOS ONE. After careful consideration, we feel that it has merit but does not fully meet PLOS ONE’s publication criteria as it currently stands. Therefore, we invite you to submit a revised version of the manuscript that addresses the points raised during the review process.

We look forward to receiving your revised manuscript.

Kind regards,

Ashraf Osman Ibrahim

Academic Editor

PLOS ONE

Journal Requirements:

4. We note that Figure 1 in your submission contain copyrighted images. All PLOS content is published under the Creative Commons Attribution License (CC BY 4.0), which means that the manuscript, images, and Supporting Information files will be freely available online, and any third party is permitted to access, download, copy, distribute, and use these materials in any way, even commercially, with proper attribution. For more information, see our copyright guidelines: http://journals.plos.org/plosone/s/licenses-and-copyright.

Reviewers' comments:

Reviewer's Responses to Questions

**Comments to the Author**

1. Is the manuscript technically sound, and do the data support the conclusions?

Reviewer #1: Partly

Reviewer #2: Yes

Reviewer #3: Yes

2. Has the statistical analysis been performed appropriately and rigorously? 

Reviewer #1: No

Reviewer #2: No

Reviewer #3: Yes

3. Have the authors made all data underlying the findings in their manuscript fully available?

Reviewer #1: Yes

Reviewer #2: No

Reviewer #3: Yes

4. Is the manuscript presented in an intelligible fashion and written in standard English?

Reviewer #1: No

Reviewer #2: Yes

Reviewer #3: Yes

5. Review Comments to the Author

Reviewer #1: 1.Incoherence in Text: Sections 2.1 and 2.2 of the paper identify some limitations of existing spectrum sensing techniques, such as sensing time, algorithm complexity, and regularization constraints. However, these limitations are not further discussed or addressed in the subsequent text. It is recommended that the article adopt a structure that separates the overview from the details to enhance coherence.

2.Lack of Innovation: The abstract and introduction provide limited information about the innovation, which seems to be a simple combination of existing CSA algorithm and DBN.

3.Confusion in Explanation: For example, the definitions of signal modification and noise alteration in Equation (3) are unclear and confusing. The definitions of H1 and H0 in (4) and (5) are not provided. The full name of RBM is not explained. The definitions of variables N and T on Page 12 are repeated and inconsistent with those on Page 8. The simulation does not provide any introduction for the benchmarks RMLSSCRN-100 and RMLSSCRN-300.

4.Lengthy Simulation with Insufficient Depth: The simulation provides a mechanical description of the experiment results without in-depth analysis. It is recommended to integrate the simulation results with the previous introduction for a more comprehensive analysis.

Reviewer #2: 1- I suggest to the authors that a native speaker review the manuscript to ensure the quality of the writing and the spelling.

2- The abstract lacks clarity and focus, requiring incorporating the underlying motivation behind the proposed methods.

3- Explicitly specifying the paper's novelty and highlighting its advancements in relation to recent state-of-the-art studies within a similar domain would add value.

4- A wide range of statistical analysis and comparison with other techniques are essential.

5- The paper could benefit from providing more detailed information about the hyperparameter selection process. Including the range of values explored, the chosen values, and the rationale behind the selection would enhance the transparency of the experimental setup and allow for better reproducibility.

6- The paper lacks details on the implementation and availability of codes. It would greatly benefit readers and future researchers to have access to the codes to replicate and facilitate further exploration of the proposed method.

7- How to run the program for the user-provided data? How to repeat the results described in the manuscript? I strongly recommend adding more detailed information, such as the saved model, the defined labels, data processing, etc. It is better to provide the documentation and sample files.

8- It is better to provide an analysis of the reason for the model's excellent performance.

9- The manuscript contains a lot of data but lacks more figures to make the results more convincing.

10- The conclusion did not summarize the improvement of the approach over other approaches and highlight what limitations the system solves. Therefore, the authors need to explain why this method has improved significantly.

11- The authors describe the methods used in detail. However, this is more literature information. It would enhance the quality of the paper if more information about the implementation and algorithms of the novel workflow were provided, too.

12- The results section needs some language revision to make it more readable.

13- You need to provide the final form of the overall loss function in the text and explain how you are weighting the different components and deciding on those weights.

14- Check that all of your Figures and Tables explain your text well.

15- In this paper, the advantages and novelty of the proposed scheme are not given.

16- It is difficult to know how to implement the proposed scheme.

Lastly, I want to emphasize that the availability of a reproducible source code. Without these components, I regret to say that I would recommend rejecting the paper.

Reviewer #3: The abstract provides a clear overview of the research on a new spectral sensing technique for cognitive radio networks (SST-CRN). Here are some suggestions:

Clarity and Conciseness:

The abstract is generally clear and concise, which is great. However, it might benefit from a slight restructuring for improved flow and readability.

Acronyms and Terminology:

Ensure that acronyms like CRN, QoS, SNR, CSA, and DBN are defined upon first use to aid readers who may not be familiar with the specific terms.

Introduction of the Problem:

Consider expanding the introduction to clearly highlight the challenges faced in spectrum sensing in cognitive radio networks. This could help set the stage for the proposed solution.

Methodology Explanation:

While you briefly mention the chicken swarm algorithm (CSA) and deep belief network (DBN), providing a bit more detail about these elements in the abstract could enhance understanding for readers.

Offline and Online Operations:

The abstract mentions the two stages of operations (offline and online), but it might be beneficial to briefly elaborate on what each stage entails. It could help readers grasp the methodology more comprehensively.

Performance Validation:

Include a sentence or two on the specific measures or metrics used for performance validation and highlight the superiority of the proposed model. It would provide a glimpse into the results and attract more interest.

Language and Style:

Consider avoiding passive voice where possible. For instance, instead of saying "The performance validation process pointed out," you might say "Performance validation results demonstrated."

Clarity and Structure:

The introduction is informative but could benefit from a clearer structure. Consider breaking it down into smaller paragraphs to enhance readability.

Definition of Terms:

While terms like spectrum sensing (SS), cyclo-stationary feature detection, and energy detection are mentioned, providing brief definitions or explanations for these terms could help readers unfamiliar with the specific terminology.

Transition to the Proposed Model:

The transition to the proposed model (SST-CRN) in the later part of the introduction is well-executed. However, consider providing a sentence that explicitly states the gap or problem in the existing methods that the proposed model aims to address.

Spectrum Sensing Techniques:

The description of various spectrum sensing techniques is informative, but it might be beneficial to briefly explain why certain techniques are preferred in specific scenarios. This could provide context for readers less familiar with the field.

Focus on Energy Detection:

The detailed explanation of energy detection and its challenges is well-presented. However, consider briefly summarizing the main challenges before introducing the proposed model.

Simulation Analyses:

It's mentioned that a wide range of simulation analyses was performed, but the abstract doesn't provide a glimpse into the results. Consider including a sentence that hints at the findings or the superiority of the proposed SST-CRN technique.

Section Organization:

The outline of the remaining sections (literature study, constraints, problem identification, modeling, reproduction findings, and conclusion) is clear. However, consider rephrasing to make it more concise.

6. PLOS authors have the option to publish the peer review history of their article (what does this mean?). If published, this will include your full peer review and any attached files.

Reviewer #1: No

Reviewer #2: No

Reviewer #3: **Yes: **BIBHAV ADHIKARI

---

## [Author Response · Author response to Decision Letter 0]

17 Mar 2024

Reviewer #1: 1. Incoherence in Text: Sections 2.1 and 2.2 of the paper identify some limitations of existing spectrum sensing techniques, such as sensing time, algorithm complexity, and regularization constraints. However, these limitations are not further discussed or addressed in the subsequent text. It is recommended that the article adopt a structure that separates the overview from the details to enhance coherence.

Response: Thanks for your valuable suggestion. We agree for the above comments. Additionally, we have included sensing time, algorithm complexity, and regularization constraints in revised manuscript to make them clear and easy to view. Refer page no 18.

One crucial parameter, sensing time, highlights how well the model can sense and adjust to dynamic changes in the radio environment. For quick decision-making and resource allocation, the least amount of sensing time must be achieved. Conversely, algorithmic complexity is a computational issue that needs to be handled with caution. It is crucial for scalability and practicality to strike a balance between the computing demands of DBN and the complexities of CSO during training and inference. Additionally, regularization requirements are essential for guaranteeing the generalization of the model and avoiding overfitting. The robust performance of the model across various CRN scenarios depends on finding the ideal balance between exploration and exploitation inside the CSO framework while respecting regularization limitations.

2.Lack of Innovation: The abstract and introduction provide limited information about the innovation, which seems to be a simple combination of existing CSA algorithm and DBN.

Response: Thanks for your valuable suggestion, we have added detail description about abstract and introduction in the revised manuscript. Refer Page No 1-5.

3.Confusion in Explanation: For example, the definitions of signal modification and noise alteration in Equation (3) are unclear and confusing. The definitions of H1 and H0 in (4) and (5) are not provided. The full name of RBM is not explained. The definitions of variables N and T on Page 12 are repeated and inconsistent with those on Page 8. The simulation does not provide any introduction for the benchmarks RMLSSCRN-100 and RMLSSCRN-300.

Response: Thanks for your valuable suggestion. We agree for the above comments. We have modified equation 3, explanation for T, N, H0 and H1 in the revised manuscript. Refer page no 10 & 11. Additionally, we have added explanation for RMLSSCRN in the result and discussion section to make them clear and easy to view.

〖 H〗_0: Y(n)=W(n),∶Primary user absent (1)

H_1:Y(n)=S(n)+W(n),∶Primary user present

Where is the example number of the sampled signal that has been received, is the sampled signal that secondary users have received, is the noise presented by the AWGN channel with zero mean and variance σ_n^2 . is the signal from PU with variance and zero mean, and h is the channel's impulse response or the channel amplitude gain between the PU transmitter and secondary user (SU) receiver since we use AWGN channel .. and stand for the alternative hypothesis (existence of the PU) and absence (null hypothesis), respectively.

〖 H〗_1: T∼Normal ((σ_n^2+σ_s^2),2N (σ_n^2+σ_s^2 )^2 ) (3)

In the equation, σ_s^2&σ_n^2 represent the signal modification and noise alteration, correspondingly. Where N is the sample number, T stands for test statistics is related to the inception (λ) for making the last decision on whether the primary signal has been current. The efficiency of energy detectors is considered three parameters existing according to the test measurements below the binary hypotheses.

4.Lengthy Simulation with Insufficient Depth: The simulation provides a mechanical description of the experiment results without in-depth analysis. It is recommended to integrate the simulation results with the previous introduction for a more comprehensive analysis.

Response: Thanks for your valuable suggestion, change has been made in the revised paper to make them clear and easy to view. Refer page no 20.

Reviewer #2: 1- I suggest to the authors that a native speaker review the manuscript to ensure the quality of the writing and the spelling.

Response: Thanks for your valuable suggestion, change has been made in the revised paper. to make them clear and easy to view.

2- The abstract lacks clarity and focus, requiring incorporating the underlying motivation behind the proposed methods.

Response: Thanks for your valuable suggestion, we have changed abstract part in the revised manuscript. Refer Page No 1.

In addition, the online spectrum sensing is carried out during the actual communication phase to adapt to dynamic changes in the spectrum environment in real-time. Offline spectrum sensing is typically performed during a dedicated sensing period before actual communication begins. The goal is to gather information about the spectral environment in advance. When combined with DBN's deep learning capabilities and CSO's innate nature-inspired algorithms, a synergistic framework is created that enables CRNs to explore and allocate frequencies on their own with astonishing accuracy. The parameters such as probability of detection, SNR of -24dB, the SST-CRN perfect has increased a developed Pd of 0.810, whereas the RMLSSCRN-100 and RMLSSCRN-300 methods have accomplished a lower Pd of 0.577 and 0.736, respectively. 

3- Explicitly specifying the paper's novelty and highlighting its advancements in relation to recent state-of-the-art studies within a similar domain would add value.

Response: Thanks for your valuable suggestion, we have added novelty and highlighting its advancements in relation to recent state-of-the-art studies in introduction part in the revised manuscript. Refer Page No 2-4.

4- A wide range of statistical analysis and comparison with other techniques are essential.

Response: Thanks for your valuable suggestion. We agree for the above comments. Additionally, we have included statistical analysis in the literature survey section in revised manuscript to make them clear and easy to view.

Table 1. A summary of related work in CRN

Authors Year Techniques Gaps or concept not covered

A. Fawzi et al. [22] 2022 Two methods ED and WD are combined to propose two-stage spectrum sensing strategies. The Shannon entropy is the only topic of this paper. It excludes Tsallis, Renyi, and Kapur entropy, among other forms of entropy.

A. D. Sahithi et al. [23] 2022 To address the issue of fading and hidden primary terminal uncertainty, cooperative spectrum sensing with ED is employed. The suggested approach is not resistant to noise improbability at low SNR, while improving SS performance. Furthermore, there is a significant increase in sensing time and computing complexity since WD and ED are merged at low SNR.

G. Prieto et al. [24] 2019 A detection technique for spectrum sensing based on entropy is suggested. A number of guidelines for figuring out how many bins a histogram has are assessed. Here are those guidelines: Scott rule, Sturges rule, and square root rule The goal of this work is not to solve the noise uncertainty issue.

F. Mashta et al. [25] 2021 The discussion of two- and three-stage SS detectors is done in detail.

Utilizing various smoothing factors, ED and maximum eigenvalue detector are employed. Because of its sensitivity to noise uncertainty, SS performs poorly overall at low SNR even with three- and two-stage spectrum sensing. Furthermore, the system's total complexity rises with the employment of the eigenvalue-based detector.

5- The paper could benefit from providing more detailed information about the hyperparameter selection process. Including the range of values explored, the chosen values, and the rationale behind the selection would enhance the transparency of the experimental setup and allow for better reproducibility.

Response: Thanks for your valuable suggestion. We have added hyperparameter selection process. Including the range of values explored, the chosen values, and the rationale behind the selection in the revised manuscript. Refer proposed section 3.1.

Fig 3. Flowchart of CSA

6- The paper lacks details on the implementation and availability of codes. It would greatly benefit readers and future researchers to have access to the codes to replicate and facilitate further exploration of the proposed method.

Response: Thanks for your valuable suggestion. We have added implementation and availability of codes in the revised manuscript. Refer proposed section.

Pseudo code of CSO

1: Prime the populace of chicken as a matrix k;

2: Determines the values of fitness for each row using the k variable; 

3: While (t<gen)

4: t = t+1; 

5: If (t %G == 0) 

6: According to their level of physical activity, 

k should be split into three groups: the rooster, the hens, and the chicks.

7: else 

8: For i = 1: 

9: If (i = rooster); 

 Upgrade the position of rooster using Eq ([Disp-formula pone.0305987.e033]); 

end if 

 10: If (i = hen); 

 Upgrade the position of hen using Eq ([Disp-formula pone.0305987.e034]); 

end if 

 11: If (i = chick); 

 Upgrade the position of chick using Eq ([Disp-formula pone.0305987.e038]); 

end if 

 12: Upgrade the new explanation of k

 13: end 

 14: end

7- How to run the program for the user-provided data? How to repeat the results described in the manuscript? I strongly recommend adding more detailed information, such as the saved model, the defined labels, data processing, etc. It is better to provide the documentation and sample files.

Response: Thanks for your valuable suggestion. We have added documentation and sample files in the revised manuscript. 

Refer this link. https://github.com/sjjana77/deep-belief-network

8- It is better to provide an analysis of the reason for the model's excellent performance.

Response: In response to the reviewer's remarks, this issue has been addressed in the revised paper.

9- The manuscript contains a lot of data but lacks more figures to make the results more convincing.

Response: Thanks for your valuable suggestion. We agree for the above comments. We have added figures 3 and 4 in the revised manuscript to make them clear and easy to view. Refer proposed method.

10- The conclusion did not summarize the improvement of the approach over other approaches and highlight what limitations the system solves. Therefore, the authors need to explain why this method has improved significantly.

Response: Thanks for your valuable suggestion. We have highlighted limitations to solve the proposed method and quantitative results in the revised manuscript. Refer conclusion section.

The limitations of proposed CSO-DBN model presents an additional degree of complexity in terms of the difficulty of reaching optimal parameter tuning because the deep learning network's parameters interact with the optimization algorithm's parameters, requiring precise calibration. Furthermore, it is still difficult to make sure the model can adapt to dynamic and unpredictable CRN environments, particularly when network circumstances are changing quickly. Scalability issues also come up, especially in large-scale CRNs where the optimization process may not be as efficient. The presented model can be tested on various SS approaches as part of the future scope. Additionally, mathematical equations can be optimized to decrease computational density and progress detection performance.

11- The authors describe the methods used in detail. However, this is more literature information. It would enhance the quality of the paper if more information about the implementation and algorithms of the novel workflow were provided, too.

Response: Thanks for your valuable suggestion. We agree for the above comments. We have added flowchart and pseudo code in the revised manuscript. 

12- The results section needs some language revision to make it more readable.

Response: In response to the reviewer's remarks, this issue has been addressed in the revised paper.

13- You need to provide the final form of the overall loss function in the text and explain how you are weighting the different components and deciding on those weights.

Response: Thanks for your valuable suggestion. We agree for the above comments. We have added overall loss function in CSO in the revised manuscript. 

All of the comparative tests in this study are conducted using Python 3.8.5 in a PC with 8GB of RAM [31]. Let us assume that the population size is 50, the maximum number of iterations is 500, the α in the fitness function is set to 0.9999, and there are 20 independent running experiments on the datasets. The selection scheme corresponding to each particle is tested for classification accuracy using the classifier. 

14- Check that all of your Figures and Tables explain your text well.

Response: In response to the reviewer's remarks, this issue has been addressed in the revised paper.

15- In this paper, the advantages and novelty of the proposed scheme are not given.

Response: Thanks for your valuable suggestion, we have added novelty and advantages of proposed method in the revised manuscript. Refer introduction section.

Energy detection is a SS model that depends on determining the absence/presence of the primary user and measuring the established signal energy by relating the selected energy level with inception. Extensive research was conducted in the survey to find the optimum threshold expression and also to develop SS performances. The researchers projected a novel technique for adaptive threshold selection in multi-band detection. Evaluating the threshold can be implemented by the function of the primary and secondary indicators of the established signals. The detection performance could be evaluated using two metrics: detection probability (DP) represents the chance that a CR user will report a PU is available when the spectra are actually being used by a PU, and false alarm probability (FA) represents the chance that a CR user will report a PU is available while the spectra are empty. Since a detection miss would demand PU intervention and an FA would lower the SE, it is typically necessary for optimal detection performance in situations where the chance of detection is increasingly susceptible to an FA probability. The effectiveness of detection in SS can be severely restricted by a number of issues, including multipath fading, shadowing, and receiver uncertainty.

16- It is difficult to know how to implement the proposed scheme.

Lastly, I want to emphasize that the availability of a reproducible source code. Without these components, I regret to say that I would recommend rejecting the paper.

Response: In response to the reviewer's remarks, this issue has been addressed in the revised paper.

Refer this link. https://github.com/sjjana77/deep-belief-network

Reviewer #3: The abstract provides a clear overview of the research on a new spectral sensing technique for cognitive radio networks (SST-CRN). Here are some suggestions:

Clarity and Conciseness:

The abstract is generally clear and concise, which is great. However, it might benefit from a slight restructuring for improved flow and readability.

Response: Thanks for your valuable suggestion, we have changed abstract part in the revised manuscript. Refer Page No 1.

In addition, the online spectrum sensing is carried out during the actual communication phase to adapt to dynamic changes in the spectrum environment in real-time. Offline spectrum sensing is typically performed during a dedicated sensing period before actual communication begins. The goal is to gather information about the spectral environment in advance. When combined with DBN's deep learning capabilities and CSO's innate nature-inspired algorithms, a synergistic framework is created that enables CRNs to explore and allocate frequencies on their own with astonishing accuracy. The parameters such as probability of detection, SNR of -24dB, the SST-CRN perfect has increased a developed Pd of 0.810, whereas the RMLSSCRN-100 and RMLSSCRN-300 methods have accomplished a lower Pd of 0.5

---

## [Decision Letter · Decision Letter 1]

16 Apr 2024

PONE-D-23-25088R1Chicken Swarm Optimization Modelling for Cognitive Radio Networks Using Deep Belief Network-Enabled Spectrum Sensing TechniquePLOS ONE

Dear Dr. M,

Thank you for submitting your manuscript to PLOS ONE. After careful consideration, we feel that it has merit but does not fully meet PLOS ONE’s publication criteria as it currently stands. Therefore, we invite you to submit a revised version of the manuscript that addresses the points raised during the review process.

We look forward to receiving your revised manuscript.

Kind regards,

Ashraf Osman Ibrahim

Academic Editor

PLOS ONE

Journal Requirements:

Reviewers' comments:

Reviewer's Responses to Questions

**Comments to the Author**

1. If the authors have adequately addressed your comments raised in a previous round of review and you feel that this manuscript is now acceptable for publication, you may indicate that here to bypass the “Comments to the Author” section, enter your conflict of interest statement in the “Confidential to Editor” section, and submit your "Accept" recommendation.

Reviewer #1: All comments have been addressed

Reviewer #2: All comments have been addressed

Reviewer #3: (No Response)

2. Is the manuscript technically sound, and do the data support the conclusions?

Reviewer #1: Yes

Reviewer #2: Yes

Reviewer #3: Partly

3. Has the statistical analysis been performed appropriately and rigorously? 

Reviewer #1: Yes

Reviewer #2: Yes

Reviewer #3: Yes

4. Have the authors made all data underlying the findings in their manuscript fully available?

Reviewer #1: Yes

Reviewer #2: Yes

Reviewer #3: Yes

5. Is the manuscript presented in an intelligible fashion and written in standard English?

Reviewer #1: Yes

Reviewer #2: Yes

Reviewer #3: Yes

6. Review Comments to the Author

Reviewer #1: (No Response)

Reviewer #2: The authors have successfully addressed this challenge with their proposed approach.

Strengths:

- The paper's organization and language are excellent and academic.

- The results are essential for transferring and fusing knowledge in the related area.

- The new proposed approach to the subject matter is significant in the novelty aspect.

Reviewer #3: The abstract effectively introduces the problem of spectrum sensing in CRNs and outlines the proposed solution. However, you could rephrase some sentences for clarity and conciseness. For example, the phrase "The proposed SST-CRN technique involves two stages of operations, namely offline and online" could be simplified to enhance readability.

The abstract presents numerical results indicating the performance of SST-CRN in terms of Pd under specific conditions (e.g., SNR of -24dB). While these results provide valuable insights, it would be beneficial to include a brief interpretation or discussion of these results to underscore the effectiveness of the proposed technique.

It could be valuable to conclude the abstract by briefly mentioning potential future directions or areas for further research, such as scalability, robustness in dynamic environments, or real-world deployment considerations.

For introduction:

The paper mentions limited research on highly dynamic CRN environments and network scaling. However, a more specific discussion on how the proposed method addresses these gaps would strengthen the argument.

While the paper acknowledges the need to explore communication overhead within the CSA framework, a dedicated section analyzing its impact, particularly in large-scale networks, is necessary.

The review mentions positive simulation results but lacks specifics. Including key metrics like detection probability, false alarm probability, and spectrum efficiency achieved by SST-CRN compared to established methods would solidify the claims of its effectiveness.

For literature review:

While summarizing existing techniques is helpful, you can include a brief comparison of their advantages and disadvantages.

Briefly explain how the reviewed methods relate to the proposed SST-CRN technique (Chicken Swarm Optimization with Deep Belief Network). Are they competing approaches, complementary methods, or foundational techniques upon which SST-CRN builds?

For the proposed model:

The section lacks information on the model's performance. Including simulation results or comparisons with existing methods would strengthen the analysis.

Briefly discuss the potential limitations of the proposed model. For example, you can mention the computational expenses involved in training the Deep Belief Network (DBN) and the potential risk of overfitting.

For Result and Discussion:

The results demonstrate that the SST-CRN model offers superior performance in spectrum sensing compared to the benchmark methods under non-fading and fading channel conditions. However, the analysis lacks information on the underlying functionalities of these benchmark methods.

For Conclusion

It restates specific results from the previous section without a broader context.

The conclusion avoids using technical jargon introduced earlier (Pd, Pe, Pnf).

It also doesn't offer a concluding statement on the overall effectiveness of the model.

Recommendations:

Briefly summarize the key performance improvements achieved by SST-CRN compared to existing methods.

Reiterate the benefits of the DBN-CSO combination without reintroducing technical terms.

Conclude by emphasizing the model's potential for improved spectrum utilization in CRNs while acknowledging the limitations for further research.

7. PLOS authors have the option to publish the peer review history of their article (what does this mean?). If published, this will include your full peer review and any attached files.

Reviewer #1: No

Reviewer #2: No

Reviewer #3: **Yes: **BIBHAV ADHIKARI

---

## [Author Response · Author response to Decision Letter 1]

6 May 2024

Reviewer #3: 

The abstract effectively introduces the problem of spectrum sensing in CRNs and outlines the proposed solution. However, you could rephrase some sentences for clarity and conciseness. For example, the phrase "The proposed SST-CRN technique involves two stages of operations, namely offline and online" could be simplified to enhance readability.

The abstract presents numerical results indicating the performance of SST-CRN in terms of Pd under specific conditions (e.g., SNR of -24dB). While these results provide valuable insights, it would be beneficial to include a brief interpretation or discussion of these results to underscore the effectiveness of the proposed technique.

It could be valuable to conclude the abstract by briefly mentioning potential future directions or areas for further research, such as scalability, robustness in dynamic environments, or real-world deployment considerations.

Response: Thanks for your valuable suggestion. We agree for the above comments. We have added detailed description about the abstract part in the revised manuscript to make them clear and easy to view. Refer page no 2.

Cognitive radio networks (CRN) enable wireless devices to sense the radio spectrum, determine the frequency state channels, and reconfigure the communication variables to satisfy Quality of Service (QoS) needs by reducing energy utilization. In CRN, spectrum sensing is an essential process that is highly challenging and can be addressed by several traditional techniques, such as energy detection, match filtering, etc. At the same time, the performance of the existing models gets affected by the low Signal to Noise Ratio (SNR) of received signals and the minimum amount of received signal samples. To resolve the drawbacks of traditional energy detection models, this paper presents a new spectral sensing technique for cognitive radio networks (SST-CRN). The proposed model helps obtain a nonlinear threshold based on a chicken swarm algorithm (CSA) with a deep belief network (DBN). The proposed DBN enabled SST-CRN technique undergo a structured process comprising two distinct stages: offline and online. In the offline stage, the DBN model is rigorously trained on pre-collected data, learning to extract complicated patterns and representations from the radio environment's spectral characteristics. This phase entails extensive feature extraction, model optimization, and validation to assure the DBN's ability to capture complex spectral dynamics. In addition, the online spectrum sensing is carried out during the actual communication phase to adapt to dynamic changes in the spectrum environment in real-time. Offline spectrum sensing is typically performed during a dedicated sensing period before actual communication begins. When combined with DBN's deep learning capabilities and CSO's innate nature-inspired algorithms, a synergistic framework is created that enables CRNs to explore and allocate frequencies on their own with astonishing accuracy. The proposed solution significantly improves the spectrum efficiency and resilience of CRNs by harnessing the power of DBN, which leads to more effective resource utilization and less interference. The Simulation results show that our proposed strategy produces more accurate spectrum occupancy assessments. The result parameters such as probability of detection, SNR of -24dB, the SST-CRN perfect has increased a developed Pd of 0.810, whereas the existing methods RMLSSCRN-100 and RMLSSCRN-300 have accomplished a lower Pd of 0.577 and 0.736, respectively. Our deep learning methodology uses convolutional neural networks to automatically learn and adapt to dynamic and complicated radio environments, improving accuracy and flexibility over classic spectrum sensing approaches. Future research might focus on improving CSO algorithms to better optimize the spectrum sensing process, enhancing the reliability of DBN-enabled sensing techniques.

For introduction:

The paper mentions limited research on highly dynamic CRN environments and network scaling. However, a more specific discussion on how the proposed method addresses these gaps would strengthen the argument.

Response: Thanks for your valuable suggestion. We have added detailed description about research gap and strengthen in the revised manuscript to make them clear and easy to view 

A potential research gap in the existing method of SST-CRN approach based on the DBN-Enabled spectrum sensing technique is the investigation of robustness and scalability. While the current strategy shows potential in improving spectrum sensing performance using deep learning techniques, further research is needed to determine its adaptability to different and dynamic CRN situations. Research might focus on analyzing the technique's performance in contexts with heterogeneous network topologies, changing interference levels, and dynamic user behaviors. Addressing the scalability issues associated with implementing DBN-based solutions in large-scale CRN deployments is another significant topic for investigation. Investigating techniques to maximize computational resources, model training efficiency, and inference speed has the potential to considerably improve the proposed technique's practical applicability and deployment feasibility in real-world CRN scenarios. To overcome this research gap would help to further our understanding and practical application of DBN-enabled SST-CRN allowing for more reliable and efficient spectrum utilization.

While the paper acknowledges the need to explore communication overhead within the CSA framework, a dedicated section analyzing its impact, particularly in large-scale networks, is necessary.

Response: Thanks for your valuable suggestion, change has been made in the revised paper to make them clear and easy to view

Recognizing the crucial role that communication overhead plays in the performance and scalability of optimization algorithms by thoroughly examining communication overhead factors such as message passing, data exchange, and synchronization mechanisms, particularly in the context of large-scale CRNs, this section aims to provide valuable insights into the efficiency and feasibility of using CSA for spectrum optimization. This investigation aims to provide realistic ideas and tactics for addressing communication overhead concerns, hence increasing the applicability and scalability of CSA-based optimization techniques in real-world CRN implementations.

The review mentions positive simulation results but lacks specifics. Including key metrics like detection probability, false alarm probability, and spectrum efficiency achieved by SST-CRN compared to established methods would solidify the claims of its effectiveness.

Response: Thanks for your valuable suggestion, change has been made in the revised paper to make them clear and easy to view

For literature review:

While summarizing existing techniques is helpful, you can include a brief comparison of their advantages and disadvantages.

Briefly explain how the reviewed methods relate to the proposed SST-CRN technique (Chicken Swarm Optimization with Deep Belief Network). Are they competing approaches, complementary methods, or foundational techniques upon which SST-CRN builds?

Response: Thanks for your valuable suggestion. We agree for the above comments change has been made in the literature section to make clear and easy to view. Refer Page no 7 to 9.

Table 1. A summary of related work in CRN

Authors Year Techniques Gaps or concept not covered Advantages Disadvantages

A. Fawzi et al. [21] 2022 Two methods ED and WD are combined to propose two-stage spectrum sensing strategies. The Shannon entropy is the only topic of this paper. It excludes Tsallis, Renyi, and Kapur entropy, among other forms of entropy. • Easy to implement

• No prior knowledge of the primary signal characteristics is required • High false alarm rate

• Unreliable at low SNR values

• Sensitive to noise uncertainty

A. D. Sahithi et al. [22] 2022 To address the issue of fading and hidden primary terminal uncertainty, cooperative spectrum sensing with ED is employed. The suggested approach is not resistant to noise improbability at low SNR, while improving SS performance. Furthermore, there is a significant increase in sensing time and computing complexity since WD and ED are merged at low SNR. • Robust against noise uncertainty

• Distinguish between signal and noise

• Decreased probability of false alarm at low SNR

 • Large sensing time to achieve a good performance

• High energy consumption when the size of the samples is large

G. Prieto et al. [23] 2019 A detection technique for spectrum sensing based on entropy is suggested. A number of guidelines for figuring out how many bins a histogram has are assessed. Here are those guidelines: Scott rule, Sturges rule, and square root rule The goal of this work is not to solve the noise uncertainty issue. • Better detection at low SNR region

• Optimal sensing

 • Prior knowledge of the primary user signal is required

• Impractical since prior knowledge about the signal is not always available

F. Mashta et al. [24] 2021 The discussion of two- and three-stage SS detectors is done in detail.

Utilizing various smoothing factors, ED and maximum eigenvalue detector are employed. Because of its sensitivity to noise uncertainty, SS performs poorly overall at low SNR even with three- and two-stage spectrum sensing. Furthermore, the system's total complexity rises with the employment of the eigenvalue-based detector. • Machine learning can detect if trained correctly can be a good approach

• Minimize the delay of the detection

 • Complex techniques

• Has to be adapted in learning in very fast changing environments

• Features selection affects detection rate and adds complexity

The existing methodologies in CRNs frequently struggle to discover dynamic spectrum possibilities in complicated and noisy situations, resulting in inefficient spectrum usage. In contrast, the proposed SST-CRN method represents a considerable divergence from existing approaches by incorporating DBNs into the spectrum sensing process. While most existing methods rely on simple signal identification algorithms, the proposed strategy uses deep learning to extract complicated patterns and representations from high-dimensional spectrum data. The proposed strategy uses DBNs to overcome the limits of traditional methods by improving spectrum sensing accuracy, flexibility, and robustness in CRNs. Thus, while previous methods set the framework for spectrum sensing in CRNs, the proposed strategy represents a transformative step forward by offering a novel approach that leverages deep learning capabilities to enable more efficient and reliable spectrum utilization.

For the proposed model:

The section lacks information on the model's performance. Including simulation results or comparisons with existing methods would strengthen the analysis.

Response: Thanks for your valuable suggestion. We agree for the above comments. We have added performance of simulations results in the revised manuscript to make clear and easy to view. 

Probability of detection: 

The possibility that a cognitive radio system can accurately detect the existence of a primary user signal through background noise or interference, permitting effective use of available spectrum resources, is known as the probability of detection.

Where:

*Q* ( *x* ) standard normal distribution's complementary cumulative distribution function is denoted as Q(x).

γ is the SNR of the received signal.

The noise power spectral density is denoted by *N* = 0.

Pnf : Probability non-fading channels refers to the possibility of meeting wireless communication channels that do not exhibit considerable fading over time during the spectrum sensing process of a cognitive radio network. This method entails finding and utilizing available spectrum opportunities in stable channels in order to improve the efficiency and reliability of spectrum usage in cognitive radio networks.

Pmd : The probability of miss alarm (Pmd) in Cognitive Radio Networks Enabled SST depicts the possibility of failing to identify the attendance of a primary user signal when it is present, indicating a potential inefficiency in spectrum sensing in cognitive radio systems.

Where:

 the likelihood of missing an alert, or going unreported, is denoted as P (miss).

N represents the total number of CR users in the network.

 Detection I is the likelihood of detection for the i-th CR user.

Probability of error: The probability of error detection is the possibility that this technique will correctly identify communication-ready spectrum bands while reducing the possibility of missing or misdetecting signals, which is essential for effective spectrum use and dependable network performance.

 is the total likelihood of error in the SST-CRN model with multiple CR users.

 is the probability of error for the i-th CR user.

Briefly discuss the potential limitations of the proposed model. For example, you can mention the computational expenses involved in training the Deep Belief Network (DBN) and the potential risk of overfitting.

Response: Thanks for your valuable suggestion. We agree for the above comments. We have added potential limitations of the proposed model in the revised manuscript to make clear and easy to view. Refer result and discussion section.

The proposed SST-CRN methods show promise for improving spectrum sensing in CRNs, but it also faces potential limitations. First challenge is the computational complexity of training and deploying deep learning models such as DBNs, which may limit real-time implementation, particularly in resource-constrained CRN devices. Furthermore, the effectiveness of the suggested technique may be significantly reliant on the availability and quality of training data, which may be limited or biased, thus compromising the model's generalization capabilities in varied CRN contexts. Furthermore, the interpretability of DBN-based spectrum sensing judgments may be problematic, making it difficult to explain and confirm the reasons behind certain detection results. Finally, while DBNs excel at learning complicated patterns, they may struggle in contexts with fast changing dynamics or adversarial conditions, requiring constant adaptation and refining to remain effective. Addressing these restrictions is critical to realizing the suggested model's full potential in real-world CRN implementations.

For Result and Discussion:

The results demonstrate that the SST-CRN model offers superior performance in spectrum sensing compared to the benchmark methods under non-fading and fading channel conditions. However, the analysis lacks information on the underlying functionalities of these benchmark methods.

Response: Thanks for your valuable suggestion. We agree for the above comments. We have added benchmark method RMLSSCRN-100 and RMLSSCRN-300 in the revised manuscript to make clear and easy to view. Refer result and discussion section.

The benchmark approaches RMLSSCRN-100 and RMLSSCRN-300 possess certain essential features that enhance their efficiency in spectrum sensing in Cognitive Radio Networks (CRNs). Both methods utilize Random Matrix Learning-based Spectrum Sensing (RMLSS) methodologies, which exploit principles from random matrix theory and machine learning algorithms. These approaches facilitate the distinction between noise and signal components in the spectrum, which is essential for precise spectrum sensing. In addition, RMLSSCRN-100 and RMLSSCRN-300 utilize sophisticated algorithms and specialized feature sets to improve the accuracy and reliability of spectrum detection. These algorithms probably include preprocessing processes to enhance raw input data, feature selection approaches to discover important spectral properties, and model modifications to enhance overall performance. Moreover, these methods may use statistical features of incoming signals, such as eigenvalue decomposition and hypothesis testing, to aid in making spectrum sensing decisions. In general, RMLSSCRN-100 and RMLSSCRN-300 exhibit advanced features designed to enhance spectrum sensing in CRNs, making them valuable standards for compa

---

## [Decision Letter · Decision Letter 2]

13 May 2024

PONE-D-23-25088R2Chicken Swarm Optimization Modelling for Cognitive Radio Networks Using Deep Belief Network-Enabled Spectrum Sensing TechniquePLOS ONE

Dear Dr. M,

Thank you for submitting your manuscript to PLOS ONE. After careful consideration, we feel that it has merit but does not fully meet PLOS ONE’s publication criteria as it currently stands. Therefore, we invite you to submit a revised version of the manuscript that addresses the points raised during the review process.

I have completed my evaluation of your manuscript. The reviewers recommend reconsideration of your manuscript following minor revisions and modifications. I invite you to resubmit your manuscript after addressing the comments below. 

When revising your manuscript, please consider all issues mentioned in the reviewers' comments carefully: Please outline every change made in response to their comments and provide suitable rebuttals for any comments not addressed. 

We look forward to receiving your revised manuscript.

Kind regards,

Ashraf Osman Ashareca

Academic Editor

PLOS ONE

Journal Requirements:

Reviewers' comments:

Reviewer's Responses to Questions

**Comments to the Author**

1. If the authors have adequately addressed your comments raised in a previous round of review and you feel that this manuscript is now acceptable for publication, you may indicate that here to bypass the “Comments to the Author” section, enter your conflict of interest statement in the “Confidential to Editor” section, and submit your "Accept" recommendation.

Reviewer #3: (No Response)

2. Is the manuscript technically sound, and do the data support the conclusions?

Reviewer #3: Yes

3. Has the statistical analysis been performed appropriately and rigorously? 

Reviewer #3: Yes

4. Have the authors made all data underlying the findings in their manuscript fully available?

Reviewer #3: Yes

5. Is the manuscript presented in an intelligible fashion and written in standard English?

Reviewer #3: Yes

6. Review Comments to the Author

Reviewer #3: Thank you for your revision. Most of the suggestions has been addressed. However there are few suggestion which I think will improve the quality of the paper:

. Introduction:

Please clearly define the research problem and objectives in the introduction to provide a more focused direction for the study.

2. Related Work:

Provide a critical analysis of the strengths and weaknesses of each reviewed technique.

Include a comparative analysis to highlight the advantages of CSO and DBN over existing methods.

3. Methodology:

It lacks clarity in explaining the selection criteria for parameters such as population size and maximum iterations. So. clearly explain the rationale behind parameter selection to justify their choices. Also, provide more detailed explanations of the CSO and DBN algorithms for readers unfamiliar with these techniques.

4. Results and Discussion:

Provide detailed analysis of the factors influencing the performance metrics, such as the impact of SNR levels on detection probabilities.

Discuss the implications of the results in terms of practical applications and theoretical advancements.

5. Conclusion:

Include a discussion on the limitations of the proposed method, such as computational complexity and real-world implementation challenges.

Provide specific directions for future research to address the identified limitations and further advance the proposed approach.

6. References:

Ensure all references include complete publication details, including journal names, volume, issue, and page numbers.

Verify the accuracy of all references to maintain the quality of the citation list.

7. PLOS authors have the option to publish the peer review history of their article (what does this mean?). If published, this will include your full peer review and any attached files.

Reviewer #3: No

---

## [Author Response · Author response to Decision Letter 2]

31 May 2024

Comments to the Author

1. If the authors have adequately addressed your comments raised in a previous round of review and you feel that this manuscript is now acceptable for publication, you may indicate that here to bypass the “Comments to the Author” section, enter your conflict of interest statement in the “Confidential to Editor” section, and submit your "Accept" recommendation.

Reviewer #3: (No Response)

2. Is the manuscript technically sound, and do the data support the conclusions?

Reviewer #3: Yes

3. Has the statistical analysis been performed appropriately and rigorously?

Reviewer #3: Yes

4. Have the authors made all data underlying the findings in their manuscript fully available?

Reviewer #3: Yes

5. Is the manuscript presented in an intelligible fashion and written in standard English?

Reviewer #3: Yes

6. Review Comments to the Author

Reviewer #3: Thank you for your revision. Most of the suggestions has been addressed. However there are few suggestion which I think will improve the quality of the paper:

. Introduction:

Please clearly define the research problem and objectives in the introduction to provide a more focused direction for the study.

Response: Thanks for your valuable suggestion, we have added research problem and objectives in the introduction section to make them clear and easy to view.

The existing methods for SST-CSO modeling in Cognitive Radio Networks (CRNs) that employ Deep Belief Network (DBN)-enabled spectrum sensing encounter problems such as large computational overhead and slow convergence speeds. These difficulties impede real-time application and efficiency in dynamic spectrum environments. Furthermore, the complexity of training DBNs and their susceptibility to parameter adjustment hinder their implementation, resulting in inefficient spectrum sensing and resource allocation in heterogeneous radio frequency environments. The proposed SST-CRN method efforts to improve the efficiency and accuracy of spectrum sensing, a fundamental task in CRNs, by exploiting CSO's powerful optimization capabilities. The integration with DBNs aims to take advantage of deep learning's ability to recognize subtle patterns and correlations in the radio frequency spectrum, which will improve channel identification. Furthermore, this hybrid method aims to lower the computational complexity and energy consumption associated with classic spectrum sensing techniques, hence increasing the operational lifespan of CRNs.

2. Related Work:

Provide a critical analysis of the strengths and weaknesses of each reviewed technique.

Include a comparative analysis to highlight the advantages of CSO and DBN over existing methods.

Response: Thanks for your valuable suggestion. We agree for the above comments. We have added strengths and weaknesses of existing system in the revised manuscript to make them clear and easy to view. Refer Literature section.

Compared to conventional optimization methods, CSO provides higher convergence rates and precision in high-dimensional search spaces, which makes it ideal for CRNs because of their complex and dynamic structure. This method, when combined with DBNs, makes use of deep learning's capacity to recognize complex relationships and patterns in spectrum data, resulting in more accurate and dependable spectrum sensing. In addition to lowering the computational load, this hybrid approach improves the overall resource allocation and utilization in CRNs by effectively balancing exploration and exploitation during optimization. Better resilience to non-stationary and heterogeneous radio environments is another feature of the CSO-DBN paradigm that guarantees more stable and interference-resistant communication. Overall, the CSO-DBN technique addresses the drawbacks of conventional approaches and satisfies the expanding requirements of contemporary wireless networks by offering a more intelligent, flexible, and effective solution for spectrum management.

3. Methodology:

It lacks clarity in explaining the selection criteria for parameters such as population size and maximum iterations. So. clearly explain the rationale behind parameter selection to justify their choices. Also, provide more detailed explanations of the CSO and DBN algorithms for readers unfamiliar with these techniques.

Response: Thanks for your valuable suggestion, change has been made in the revised paper to make them clear and easy to view.

To achieve optimal performance, it is important to set parameters for CSO in CRN using Deep Belief Network (DBN)-enabled spectrum sensing. In CSO, the population size must strike a balance between computing efficiency and diversity. A population size of 20–50 is often selected to allow for a thorough search space exploration while maintaining reasonable processing needs. Another crucial parameter is the maximum number of iterations, which is often set between 100 and 500 iterations to provide the algorithm enough time to converge to an ideal solution without creating undue delay a crucial factor in contexts with dynamic spectrum. Furthermore, CSO-specific characteristics, including the quantity of leaders and followers, are set up to preserve a balance between the search space's exploration and exploitation. The complexity of the CRN environment and the available computational resources are taken into consideration when fine-tuning these settings, which are tested empirically. The meticulous adjustment of parameters guarantees that the CSO algorithm combines with the DBN in an efficient and precise manner for spectrum sensing, hence improving the overall performance of the network.

Chicken Swarm Optimization (CSO) is a nature-inspired optimization algorithm modeled after the social behaviours and hierarchical structures of chicken swarms. In CSO, the population consists of roosters, hens, and chicks, each playing a distinct role in the optimization process. Roosters, representing the best solutions, guide the hens and chicks, which represent other candidate solutions. The algorithm iterates through movements and interactions among these groups to explore and exploit the search space effectively, aiming to find the optimal solution. Deep Belief Networks (DBNs) are a type of deep learning model composed of multiple layers of Restricted Boltzmann Machines (RBMs), which are simple, two-layer neural networks. DBNs are trained in a greedy layer-wise manner, starting with the first RBM and progressively training subsequent layers while keeping the previously trained layers fixed. This hierarchical learning approach allows DBNs to capture complex patterns and dependencies in data. In spectrum sensing for Cognitive Radio Networks (CRNs), DBNs are used to process and analyse the radio frequency spectrum data, identifying available channels with high accuracy. By combining CSO and DBN, we leverage CSO's optimization capabilities to fine-tune the parameters of the DBN, enhancing its performance in spectrum sensing. This synergy ensures efficient spectrum utilization and robust detection of available channels in dynamic wireless environments.

4. Results and Discussion:

Provide detailed analysis of the factors influencing the performance metrics, such as the impact of SNR levels on detection probabilities.

Discuss the implications of the results in terms of practical applications and theoretical advancements.

Response: Thanks for your valuable suggestion. We agree for the above comments. We have added performance metrics, such as the impact of SNR levels on detection probabilities in the revised manuscript to make them clear and easy to view and also included implications of the results in terms of practical applications and theoretical advancements in the revised manuscript.

The impact of SNR values on detection probability in spectrum sensing is significant. At high SNR levels, the primary user's signal is significantly stronger than the noise, resulting in higher detection probability and lower false alarm rates due to the clear separation between signal and noise. In contrast, at low SNR levels, the signal is closer in magnitude to the noise, making it harder to discern between the two. This leads to reduced detection probabilities and greater false alarm rates since the sensing algorithm fails to effectively recognize the presence of the primary user among the noise. The performance of spectrum sensing techniques is heavily dependent on SNR levels, with high SNR resulting in higher detection probability and low SNR providing considerable obstacles. Other important elements that influence detection performance include sensing time, channel conditions, noise characteristics, and signal qualities. Optimizing these aspects using modern algorithms and adaptive sensing approaches can considerably improve the reliability and efficiency of spectrum sensing in cognitive radio networks.

In practically, it leads to more efficient, reliable, and adaptable spectrum sensing in cognitive radio networks, which improves overall communication quality and device performance. Theoretically, it stimulates the development of sophisticated algorithms and models that push the limits of present spectrum sensing capabilities, driving innovation and growth in wireless communications.

5. Conclusion:

Include a discussion on the limitations of the proposed method, such as computational complexity and real-world implementation challenges.

Provide specific directions for future research to address the identified limitations and further advance the proposed approach.

Response: Thanks for your valuable suggestion, we have added detail description about limitations of proposed method and future research in the revised manuscript. 

The major limitation of the proposed method SST-CRN is the high computational complexity and resource requirements of training and deploying DBNs, which can be difficult in situations when resources are limited or real-time. Furthermore, despite its effectiveness in optimization, the CSO algorithm can occasionally experience problems with convergence speed and become stuck in local optima, particularly in contexts with a highly dynamic and heterogeneous spectrum.

In Future research could focus on creating more efficient and scalable versions of CSO in order to improve its convergence features and reduce computational overhead. To address local optima issues, hybrid approaches combining CSO with other optimization techniques may be investigated. Furthermore, advances in lightweight and real-time deep learning models may make DBN-enabled spectrum sensing more useful for CRNs. Adaptive learning techniques could also be investigated, allowing DBNs to continuously learn and adapt to different spectrum settings with less input. Finally, incorporating modern data fusion algorithms to collect and process a variety of spectrum sensing data sources could increase the technique's dependability and accuracy.

6. References:

Ensure all references include complete publication details, including journal names, volume, issue, and page numbers.

Verify the accuracy of all references to maintain the quality of the citation list.

Response: Thanks for your valuable suggestion. We agree for the above comments. We have added below references in the revised manuscript to make them clear and easy to view. Refer Literature section.

---

## [Decision Letter · Decision Letter 3]

10 Jun 2024

Chicken Swarm Optimization Modelling for Cognitive Radio Networks Using Deep Belief Network-Enabled Spectrum Sensing Technique

PONE-D-23-25088R3

Dear Dr. Saraswathi,

We’re pleased to inform you that your manuscript has been judged scientifically suitable for publication and will be formally accepted for publication once it meets all outstanding technical requirements.

Kind regards,

Ashraf Osman 

Academic Editor

PLOS ONE

Additional Editor Comments (optional):

Reviewers' comments:

Reviewer's Responses to Questions

**Comments to the Author**

1. If the authors have adequately addressed your comments raised in a previous round of review and you feel that this manuscript is now acceptable for publication, you may indicate that here to bypass the “Comments to the Author” section, enter your conflict of interest statement in the “Confidential to Editor” section, and submit your "Accept" recommendation.

Reviewer #3: All comments have been addressed

2. Is the manuscript technically sound, and do the data support the conclusions?

Reviewer #3: Yes

3. Has the statistical analysis been performed appropriately and rigorously? 

Reviewer #3: Yes

4. Have the authors made all data underlying the findings in their manuscript fully available?

Reviewer #3: Yes

5. Is the manuscript presented in an intelligible fashion and written in standard English?

Reviewer #3: Yes

6. Review Comments to the Author

Reviewer #3: Well Done. It has addressed all the queries and comments previously raised. Thank you for the opportunity.

7. PLOS authors have the option to publish the peer review history of their article (what does this mean?). If published, this will include your full peer review and any attached files.

Reviewer #3: No

---

## [Editor Report · Acceptance letter]

20 Jun 2024

PONE-D-23-25088R3 

PLOS ONE

Dear Dr. M, 

I'm pleased to inform you that your manuscript has been deemed suitable for publication in PLOS ONE. Congratulations! Your manuscript is now being handed over to our production team.

Kind regards, 

on behalf of

Dr. Ashraf Osman Ashareca 

Academic Editor

PLOS ONE